

# The SPARC water vapour assessment II:
# Comparison of stratospheric and lower mesospheric water vapour time series observed from satellites

Farahnaz Khosrawi[1], Stefan Lossow[1], Gabriele P. Stiller[1], Karen H. Rosenlof[2], Joachim Urban[3,†], John P. Burrows[4], Robert P. Damadeo[5], Patrick Eriksson[3], Maya García-Comas[6], John C. Gille[7,8], Yasuko Kasai[9], Michael Kiefer[1], Gerald E. Nedoluha[10], Stefan Noël[4], Piera Raspollini[11], William G. Read[12], Alexei Rozanov[4], Christopher E. Sioris[13], Kaley A. Walker[14], and Katja Weigel[4]

[1]Karlsruhe Institute of Technology, Institute of Meteorology and Climate Research, Hermann-von-Helmholtz-Platz 1, 76344 Eggenstein-Leopoldshafen, Germany
[2]NOAA Earth System Research Laboratory, Global Monitoring Division, 325 Broadway, Boulder, CO 80305, USA
[3]Chalmers University of Technology, Department of Space, Earth and Environment, Hörsalsvägen 11, 41296 Göteborg, Sweden
[4]University of Bremen, Institute of Environmental Physics, Otto-Hahn-Allee 1, 28334 Bremen, Germany
[5]NASA Langley Research Center, Mail Stop 401B, Hampton, VA 23681, USA
[6]Instituto de Astrofísica de Andalucía (IAA-CSIC), Glorieta de la Astronomía, 18008 Granada, Spain
[7]National Center for Atmospheric Research, Atmospheric Chemistry Obserations and Modeling Laboratory, P.O. Box 3000, Boulder, CO 80307-3000, USA
[8]University of Colorado, Atmospheric and Oceanic Sciences, Boulder, CO 80309-0311, USA
[9]National Institute of Information and Communications Technology, Terahertz Technology Research Center, 4-2-1 Nukui-kita, Koganei, Tokyo 184-8795, Japan
[10]Naval Research Laboratory, Remote Sensing Devision, 4555 Overlook Avenue Southwest, Washington, DC 20375, USA
[11]Istituto di Fisica Applicata del Consiglio Nazionale delle Ricerche (IFAC-CNR), Via Madonna del Piano, 10, 50019 Sesto Fiorentino, Italy
[12]Jet Propulsion Laboratory, 4800 Oak Grove Drive, Pasadena, CA 91109, USA
[13]Environment and Climate Change Canada, Atmospheric Science and Technology Directorate, 4905 Dufferin St., Toronto, ON, M3H 5T4, Canada
[14]University of Toronto, Department of Physics, 60 St. George Street, Toronto, ON, M5S 1A7, Canada
[†]deceased, 14 August 2014

*Correspondence to:* Farahnaz Khosrawi (farahnaz.khosrawi@kit.edu)

**Abstract.** Time series of stratospheric and lower mesospheric water vapour using 33 data sets from 15 different satellite instruments were compared in the framework of the second SPARC (Stratosphere-troposphere Processes And their Role in Climate) water vapour assessment (WAVAS-II). This comparison aimed to provide a comprehensive overview of the typical uncertainties in the observational database that can be considered in the future in observational and modelling studies addressing e.g stratospheric water vapour trends. The time series comparisons are presented for the three latitude bands, the Antarctic (80°–70°S), the tropics (15°S–15°N) and the northern hemisphere mid-latitudes (50°–60°N) at four different altitudes (0.1, 3, 10 and 80 hPa) covering the stratosphere and lower mesosphere. The combined temporal coverage of observations from the 15



satellite instruments allowed considering the time period 1986–2014. In addition to the qualitative comparison of the time series, the agreement of the data sets is assessed quantitatively in the form of the spread (i.e. the difference between the maximum and minimum volume mixing ratio among the data sets), the (Pearson) correlation coefficient and the drift (i.e. linear changes of the difference between time series over time). Generally, good agreement between the time series was found in the middle stratosphere while larger differences were found in the lower mesosphere and near the tropopause. Concerning the latitude bands, the largest differences were found in the Antarctic while the best agreement was found for the tropics. From our assessment we find that all data sets can be considered in the future in observational and modelling studies addressing e.g. stratospheric and lower mesospheric water vapour variability and trends when data set specific characteristics (e.g. a drift) and restrictions (e.g. temporal and spatial coverage) are taken into account.

## 1   Introduction

Water vapour is the most important greenhouse gas and plays a key role in the chemistry and radiative balance of the atmosphere. Any changes in atmospheric water vapour have important implications for the global climate (Solomon et al., 2010; Riese et al., 2012) and need to be monitored and understood (Müller et al., 2016). Accurate knowledge of the water vapour distribution and its trends from the upper troposphere up to the mesosphere is therefore crucial for understanding climate change and chemical forcing (Hegglin et al., 2013).

Water vapour is the source of the hydroxyl radical (OH) which controls the lifetime of shorter-lived pollutants, tropospheric and stratospheric ozone and other longer-lived greenhouse gases such as methane (Seinfeld and Pandis, 2006). Further, water vapour is an essential component of Polar Stratospheric Clouds (PSCs) which play a key role in Antarctic and Arctic ozone depletion during winter and spring (Solomon, 1999). Accordingly, water vapour has an important influence on stratospheric chemistry through its ability to form ice particles. Dehydration, that is, the removal of water vapour from the gas phase, can either be a reversible or an irreversible process depending on the lifetime of water-containing particles and their size. However, ice particles generally live long enough and grow sufficiently large to fall and remove water vapour permanently from an air mass so that dehydration can generally be defined as an irreversible process. Dehydration in the stratosphere is generally observed over the Antarctic during winter (e.g. Kelly et al., 1989; Vömel et al., 1995; Nedoluha et al., 2000, 2007) and to a lesser extent also over the Arctic (e.g. Fahey et al., 1990; Pan et al., 2002; Khaykin et al., 2013; Manney and Lawrence, 2016) as well as at the tropical tropopause (e.g. Jensen et al., 1996; Read et al., 2004; Schiller et al., 2009).

In addition to its role in the Earth's radiative budget and middle atmospheric chemistry, water vapour is an important tracer for transport in the stratosphere and lower mesosphere. Dynamical circulations that can be diagnosed with water vapour in the middle atmosphere are the Brewer–Dobson circulation in the stratosphere and the pole-to-pole circulation in the mesosphere (Brewer, 1949; Remsberg et al., 1984; Mote et al., 1996; Pumphrey and Harwood, 1997; Seele and Hartogh, 1999; Lossow et al., 2017a). In the stratosphere, the water vapour abundance is primarily governed by two main sources: (1) the transport from the troposphere through the tropical tropopause layer (TTL), where the minimum temperature (the so-called cold point temperature) determines how much water vapour enters the stratosphere (Fueglistaler and Haynes, 2005). (2) the oxidation of



methane, which is the only important chemical source of water vapour in the stratosphere (Bates and Nicolet, 1950; Le Texier et al., 1988).

A major research focus in relation to water vapour has been on the detection and attribution of long-term changes in stratospheric and mesospheric water vapour based on in-situ and remote sensing measurements (Oltmans and Hofmann, 1995; Oltmans et al., 2000; Rosenlof et al., 2001; Nedoluha et al., 2003; Scherer et al., 2008; Hurst et al., 2011; Hegglin et al., 2014; Dessler et al., 2014). Many of these measurements have indicated an increase in stratospheric and mesospheric water vapour that has significant implications for atmospheric temperature. Increases in stratospheric water vapour cool the stratosphere but warm the troposphere (Solomon et al., 2010). Model simulations predict a $\sim 1\,\mathrm{K}$ decrease in stratospheric temperature per decade along with a 0.5–1 ppmv increase of water vapour in the 21st century (Gettelman et al., 2010). Both the future cooling of the stratosphere and the future increase in water vapour enhance the potential for the formation of PSCs which would have significant implications on Arctic and Antarctic dehydration and ozone loss (Khosrawi et al., 2016; Thölix et al., 2016). The methane increase in the stratosphere can only explain part of the observed water vapour changes (e.g. Rosenlof et al., 2001; Hurst et al., 2011). A complete understanding of water vapour changes also requires good knowledge of short-term variability, such as the annual and semi-annual variation or the variations caused by the quasi-biennial oscillation (e.g. Schoeberl et al., 2008; Remsberg, 2010; Kawatani et al., 2014; Lossow et al., 2017b).

In addition to an observed long-term increase in stratospheric water vapour, pronounced drops have occasionally been observed. One drop (also known as the millennium drop) occurred in 2000 (Randel et al., 2006; Scherer et al., 2008; Solomon et al., 2010; Urban et al., 2012; Brinkop et al., 2016), where the water vapour volume mixing ratios first started to recover in 2004 to 2005. This decrease was caused by a reduced transport of water vapour across the tropical tropopause in response to lower cold point temperatures. The exact driving mechanism is still in question, but has been suggested to be due to variations of the QBO (quasi-biennial oscillation), ENSO (El Niño Southern Oscillation) and the Brewer-Dobson circulation that collectively acted in the same direction lowering the tropopause temperatures. In 2011 and 2012 another drop occurred, which however was more short-lived than the millennium drop (Urban et al., 2014). Recently, another sharp decrease was observed in connection with the QBO disruption and the unusual El Niño event in 2015 and 2016 (Tweedy et al., 2017; Avery et al., 2017), but also this one has already recovered.

Within the framework of the second SPARC water vapour assessment (WAVAS-II), we compared time series of stratospheric and lower mesospheric water vapour derived from a number of different satellite data sets. The time series comparison was performed for the Antarctic (80°–70°S), the tropics (15°S–15°N) and the northern hemisphere mid-latitudes (50°–60°N) at four different altitudes (0.1, 3, 10 and 80 hPa). This selection of latitude bands covers all three basic climatic regions (i.e. tropics, mid-latitudes and polar region) and allows the inclusion of all stratospheric WAVAS-II data sets in the comparison. The combined temporal coverage of the 15 satellite instruments allows the consideration of the time period 1986–2014. This work aims to provide estimates of the typical uncertainties in the time series from satellite observations that should be taken into account in observational and modelling studies. A brief overview of the data sets used in this study is provided in the next section followed by a description of the analysis approach in Sect. 3. In Section 4 the results are presented, focusing on the



comparison of the de-seasonalised water vapour time series. Comparison results for the absolute time series are given in the Supplement. Finally, our results will be summarised and conclusions will be given in Sect. 5.

## 2 Data sets

For the comparison of water vapour products performed within the second SPARC WAVAS-II assessment, 40 data sets (not
5 including data sets of minor water vapour isotopologues) have been considered, primarily focusing on the time period from 2000 to 2014 (Walker and Stiller, in preparation). In the present study, we included all 33 data sets that have observational coverage in the stratosphere. A list of these data sets is provided in Table 1, along with the effective time periods available for analysis. In addition, this table provides the data sets labels and numbers used in the figures. Overall, data sets from the following 15 instruments have been considered (listed in alphabetical order): ACE-FTS, GOMOS, HALOE, HIRDLS, ILAS-II,
10 MAESTRO, MIPAS, MLS (aboard the Aura satellite, not the instrument on the Upper Atmosphere Research Satellite – UARS), POAM III, SAGE II, SAGE III, SCIAMACHY, SMILES, SMR and SOFIE. For a number of instruments there are multiple data sets based on different data processors, measurement geometries, retrieval versions and spectral signatures used to derive the water vapour information. The HALOE, POAM III and SAGE II data sets also include observations before 2000. These were considered in the comparisons, so that the combined temporal coverage of all data sets ranges from 1986 to 2014. A
15 complete description of the data sets and their characteristics can be found in the WAVAS-II data set overview paper by Walker and Stiller (in preparation). In comparison to our previous SPARC WAVAS-II paper (Lossow et al., 2017b) the following two data related changes have been made: (1) The ACE-FTS v3.5 and MAESTRO data sets have been extended from March 2013 until December 2014 (see Tab. 1 of Lossow et al., 2017b). (2) The MIPAS ESA v7 data set has become complete. Previously, this data sets comprised only a sample of 200000 observations (instead of 1800000), however the temporal coverage on a
20 monthly basis was already complete.

## 3 Approach

### 3.1 Time series calculation

For the first step, we screened the individual data sets according to the criteria recommended by the data providers. A complete list of these criteria is given in the WAVAS-II data set overview paper by Walker and Stiller (in preparation). After the screening
25 we interpolated the data onto a regular pressure grid. This comprises 32 levels per pressure decade, which corresponds to a fine vertical sampling of about 0.5 km. The uppermost level we consider is 0.1 hPa. The interpolated profiles were then binned monthly and for three latitude bands chosen: 80°–70°S, 15°S–15°N and 50°–60°N. The monthly zonal means $y_a(t, \phi, z)$ are given as:

$$y_a(t,\phi,z) = \frac{1}{n_o(t,\phi,z)} \sum_{i=1}^{n_o(t,\phi,z)} x_i(t,\phi,z). \tag{1}$$





In the equation above $x_i(t,\phi,z)$ describes the individual observations that fall into a given time $t$ (i.e. month) and latitude $\phi$ bin, $n_o(t,\phi,z)$ indicates their total number and $z$ denotes the altitude level. Before this calculation the data in the given bin were screened using the median and the median absolute difference (MAD, Jones et al., 2012) in an attempt to remove unrepresentative observations that occasionally occur. Data points outside the interval $\langle\text{median}[x_i(t,\phi,z)] \pm 7.5\ \text{MAD}[x_i(t,\phi,z)]\rangle$,

5   with $i = 1, ..., n_o(t,\phi,z)$, were discarded, targeting the most prominent outliers (Jones et al., 2012; Lossow et al., 2017b). For a normally distributed data set, 7.5 MAD corresponds to about $5\sigma$. For individual data sets this concerned on average between 0.03% and 3.2% percent of the data in a given bin. Averaged over all data sets typically 0.6% of the data in a given bin were removed by this screening. In addition to the monthly zonal means, the corresponding standard error $\epsilon_a(t,\phi,z)$ was calculated by:

$$\epsilon_a(t,\phi,z) = \sqrt{\frac{1}{n_o(t,\phi,z)[n_o(t,\phi,z)-1]} \sum_{i=1}^{n_o(t,\phi,z)} [x_i(t,\phi,z) - y_a(t,\phi,z)]^2}. \quad (2)$$

To avoid spurious data, averages that are smaller than their corresponding standard errors in an absolute scale are discarded. Also, monthly averages based on less than 20 observations for dense data sets (e.g. HIRDLS, MIPAS, MLS, SCIAMACHY limb, SMILES-NICT and SMR) and less than 5 observations for sparse data sets (e.g. ACE-FTS, GOMOS, HALOE, ILAS-II, MAESTRO, POAM III, SAGE II, SAGE III, SCIAMACHY occultation and SOFIE) were not considered any further.

15  This is a slightly more refined approach than used in the time series analysis by Lossow et al. (2017b) where a minimum of 20 observations was required for all data sets. However, additional tests have shown that such a conservative criterion is not required for the sparser data sets.

In our analysis we consider both absolute time series and de-seasonalised time series. The ILAS-II and SMILES data sets cover less than one year, so that a de-seasonalisation is not meaningful. There are multiple ways to achieve a de-seasonalisation.

20  The most common and simplest approach is to calculate for a given calendar month the average over several years. Subsequently this average is subtracted from the individual months contributing to this climatological average (aka average approach). This approach requires that a data set covers every calendar month at least twice. For the MIPAS V5H data sets this requirement is not fulfilled as they cover only 21 months. To accomplish a de-seasonalisation even for these data sets a regression approach was used. Every data set was regressed with the following regression model:

$$\begin{aligned} f(t,\phi,z) = {} & C_{\text{offset}}(\phi,z) + \\ & C_{\text{AO}_1}(\phi,z) \cdot \sin(2\pi t/p_{\text{AO}}) + C_{\text{AO}_2}(\phi,z) \cdot \cos(2\pi t/p_{\text{AO}}) + \\ & C_{\text{SAO}_1}(\phi,z) \cdot \sin(2\pi t/p_{\text{SAO}}) + C_{\text{SAO}_2}(\phi,z) \cdot \cos(2\pi t/p_{\text{SAO}}). \end{aligned} \quad (3)$$

This model contained an offset as well as the annual (AO) and semi-annual variation (SAO). The AO and SAO are parameterised by orthogonal sine and cosine functions. $f(t,\phi,z)$ denotes the fit of the regressed time series and $C$ are the regression coefficients of the individual model components. $p_{\text{AO}}=1$ year is the period of the annual variation, likewise $p_{\text{SAO}}=0.5$ years





is the period of the semi-annual variation. In accordance to $p_{\mathrm{AO}}$ and $p_{\mathrm{SAO}}$ given in years, the time $t$ is here also used in a yearly scale. To calculate the regression coefficients we followed the method outlined by von Clarmann et al. (2010) using the standard errors $\epsilon_a(t,\phi,z)$ (their inverse squared) of the monthly zonal means as statistical weights. Autocorrelation effects and empirical errors (Stiller et al., 2012) were not considered in this regression. The de-seasonalised time series $y_d(t,\phi,z)$, thus the anomalies for each time t, are then given as:

$$y_d(t,\phi,z) = y_a(t,\phi,z) - f(t,\phi,z). \tag{4}$$

For the sake of simplicity we do not assign any error to the regression fit, so that the standard error of the de-seasonalised time series is given by:

$$\epsilon_d(t,\phi,z) = \epsilon_a(t,\phi,z). \tag{5}$$

## 3.2 Comparison parameters

To assess how the different time series compare between two data sets or alltogether we use a number of parameters, namely the spread (i.e. the difference between the maximum and minimum volume mixing ratio among the data sets), the (Pearson) correlation coefficient and the drift (i.e. linear changes of the difference between time series over time). In the following subsections, the calculation of these parameters is described in more detail.

### 3.2.1 Spread

We define the spread as the difference between the maximum and minimum volume mixing ratio among the data sets at a given time and place. As such, the spread is a simple measure of the collective consistency among the time series from the different data sets. We have chosen this approach for the spread calculation since for the other approaches based on standard deviation or percentiles, assumptions have to be made. However, we have also calculated the spread using the other two approaches and derived qualitatively the same results as for the maximum-minimum calculation. Prior to the spread calculation, we performed an additional screening among the data sets to avoid unrepresentative spread estimates. The screening is again based on the median and median absolute difference, as done before for the the monthly zonal mean calculation. Monthly zonal means outside the interval $\langle \mathrm{median}[y_p(t,\phi,z)_i] \pm 7.5\,\mathrm{MAD}[y_p(t,\phi,z)_i]\rangle$ were not considered, with $i=1,...,n_d(t,\phi,z)$ and $n_d(t,\phi,z)$ denoting the number of data sets at a given time, latitude and altitude. The subscript $p$ is used as a placeholder either for the absolute or the de-seasonalised data. This screening removed overall 2.6% of the data for the latitude band between $80°$ and $70°$S. For the tropical and the mid-latitude bands it was 3.6% and 3.7%, respectively. Subsequently, the spread was derived. We did not impose any additional criterion on the number of data sets available for a spread estimate to be valid (two data sets is the natural minimum). However, for much of the 1990s the only available satellite data sets are HALOE and SAGE II. Since both instruments provide solar occultation measurements the number of coincidences is limited. Thus, their time series do not





constantly overlap, there are many gaps in the spread. Therefore, we focus in the result section on the time period between 2000 and 2014.

### 3.2.2 Correlation

To describe the consistency between two time series we employed the correlation coefficient $r(\phi, z)$:

$$r(\phi,z) = \frac{\sum_{i=1}^{n_t(\phi,z)} [y_p(t_i,\phi,z)_1 - \overline{y_p}(\phi,z)_1] \cdot [y_p(t_i,\phi,z)_2 - \overline{y_p}(\phi,z)_2]}{\sqrt{\sum_{i=1}^{n_t(\phi,z)} \cdot [y_p(t_i,\phi,z)_1 - \overline{y_p}(\phi,z)_1]^2} \cdot \sqrt{\sum_{i=1}^{n_t(\phi,z)} [y_p(t_i,\phi,z)_2 - \overline{y_p}(\phi,z)_2]^2}} \tag{6}$$

with

$$\overline{y_p}(\phi,z)_1 = \frac{1}{n_t(\phi,z)} \sum_{i=1}^{n_t(\phi,z)} y_p(t_i,\phi,z)_1 \quad \text{and} \tag{7}$$

$$\overline{y_p}(\phi,z)_2 = \frac{1}{n_t(\phi,z)} \sum_{i=1}^{n_t(\phi,z)} y_p(t_i,\phi,z)_2. \tag{8}$$

The subscripts at the end of the variables refer to the two data sets. $p$ is again a placeholder for the absolute and de-seasonalised data. $n_t(\phi, z)$ is the number of months the two time series actually overlap, i.e. where both data sets yield valid monthly means. Correlation coefficients were only considered if the overlap was at least 12 months. We did not perform any significance analysis for the coefficients since we simply want to show if the expected high correlation between two time series exist.

### 3.3 Drift

As drift we consider the linear change of the difference between two time series, which indicates if the longer-term variation of the two time series is the same or not. The difference time series was calculated as:

$$\Delta y_d(t,\phi,z) = y_d(t,\phi,z)_1 - y_d(t,\phi,z)_2, \tag{9}$$

where the subscripts at the end once more denote the two data sets. As indicated by this equation the drift analysis focuses on de-seasonalised time series. The standard error corresponding to the difference time series is given by:

$$\Delta\epsilon_d(t,\phi,z) = \sqrt{\epsilon_d(t,\phi,z)_1^2 + \epsilon_d(t,\phi,z)_2^2}. \tag{10}$$

Due to the lack of appropriate covariance data, this calculation omits any covariance between the different data sets. The difference time series were then regressed with a regression model containing an offset, a linear term (which describes the



drift) and the QBO parameterised by the Singapore (1°N, 104°E) winds at 50 hPa ($QBO_1$) and 30 hPa ($QBO_2$) provided by Freie Universität Berlin (http://www.geo.fu-berlin.de/met/ag/strat/produkte/qbo/qbo.dat):

$$f(t, \phi, z) = C_{\text{offset}}(\phi, z) + C_{\text{linear}}(\phi, z)t +$$
$$C_{QBO_1}(\phi, z) \cdot QBO_1(t) + C_{QBO_2}(\phi, z) \cdot QBO_2(t). \tag{11}$$

The calculation of the regression coefficients followed again the method by von Clarmann et al. (2010), using the inverse square of the corresponding standard error $\Delta\epsilon_d(t, \phi, z)$ as weight. Here, unlike in the regression for the de-seasonalisation, auto-correlation effects and empirical errors were considered to derive optimal uncertainty estimates for the drifts. This consideration used the approach outlined by Stiller et al. (2012). We show drift results if the overlap period between the two time series is at least 36 months. As overlap period we define the time between the first and the last month both data sets yield a valid monthly mean. We also provide the information regarding how many months both data sets actually overlap, but we did not put any additional constraint on this quantity. In addition, we have performed tests with more advanced regression models, which yielded qualitatively the same results.

## 4 Results

In this section, the results for the time series comparison are presented. First, we provide an example (Fig. 1) of the typical altitude-time distribution (contour time series) to describe the general characteristics of the water vapour distribution in the three latitude bands considered: Antarctic (80°–70°S), tropics (15°S–15°N) and the northern hemisphere mid-latitudes (50°–60°N). These latitude bands were selected since these cover all three basic climatic regions and allow the inclusion of all stratospheric WAVAS-II data sets in the comparison. Contour time series of water vapour in these three latitude bands derived from all of the data sets considered in this study are provided in the Supplement (Fig. S1-S3). These figures give a good first overview of the altitude and temporal coverage of the individual data sets and their representation of the characteristics of the water vapour distribution at the three latitude bands.

The comparison of the time series is then performed qualitatively for all data sets at the three latitude bands and at four selected altitudes covering the stratosphere and lower mesosphere (0.1, 3, 10 and 80 hPa). Subsequently, we assess the agreement of the data sets quantitatively in form of the spread over all data sets as well as the correlations and drifts among the individual data sets. While the example is based on absolute data, the comparison results presented in this section are derived from de-seasonalised data. The corresponding results based on absolute data (except for the drift) are provided in the Supplement.

### 4.1 General characteristics of the water vapour time series

Figure 1 shows contour time series of water vapour in the Antarctic (80°–70°S), tropics (15°S–15°N) and mid-latitudes (50°–60°N) based on the MLS data set for the time period 2004–2014. Here, the typical characteristics of the water vapour distributions in these latitude regions become visible. The water vapour distribution in the polar regions (Fig. 1 top) is determined



by the following three processes (1) dehydration of the lower stratosphere during polar winter caused by the sedimentation of ice containing polar stratospheric cloud particles (Kelly et al., 1989; Fahey et al., 1990), (2) vertical transport of dry/moist air. During polar winter, dry air from the upper mesosphere descends within the polar vortex, while during summer and early autumn moist air from the upper stratosphere is transported into the lower mesosphere and (3) enhanced production of water vapour by methane oxidation during summer due to the higher insolation (Bates and Nicolet, 1950; Le Texier et al., 1988).

In the tropics (Fig. 1 middle), the most prominent feature in the water vapour time series is the "atmospheric tape recorder" (Mote et al., 1996). This feature is a consequence of the annual variation of dehydration (or freeze-drying) at the tropical tropopause due to the annual variation of the tropical tropopause temperature. The tape recorder signal is transported upwards to about $15\,\mathrm{hPa}$ by the ascending branch of the Brewer-Dobson circulation and maintains its integrity because of the subtropical mixing barrier in the lower stratosphere. Around the stratopause ($\sim 1\,\mathrm{hPa}$) a pronounced semi-annual variation is found that is induced by an interplay of transport and momentum deposition of different types of waves (Hamilton, 1998).

The water vapour distribution in the mid-latitudes (Fig. 1 bottom) is primarily influenced by transport within the Brewer-Dobson circulation and the overturning circulation in the mesosphere. In the lower stratosphere, low volume mixing ratios are transported from the lower latitudes to the mid-latitudes in late spring/early summer (Ploeger et al., 2013). Likewise, in the lower mesosphere the effect of upwelling in summer and downwelling in winter can be clearly seen, as described for the Antarctic.

## 4.2 Qualitative time series comparisons

In the following, the time series from the different satellite data sets are qualitatively compared. The time series in the three considered latitudes bands cover generally the time period from 1991–2014 ($0.1\,\mathrm{hPa}$), from 1986 to 2014 (3 and $10\,\mathrm{hPa}$) and 1988–2014 ($80\,\mathrm{hPa}$). A necessary requirement for the analyses of the de-seasonalised time series was a minimum data set length of one year, ruling out some shorter data sets (see Sect. 3.1). However, these data sets are considered in the Supplement where the time series in absolute terms derived from all satellite instruments considered in this study are provided (Fig. S3–S6). Some data sets as e.g. the MAESTRO data set only have coverage up to the lowest pressure level ($80\,\mathrm{hPa}$) considered here and thus these data can only be found in bottom subfigures (Fig. 2–Fig. 4 and Fig S3–S6). Overall, 25 data sets have been considered in the comparison for the Antarctic while 24 data sets have been considered in the comparison for the tropics. In the northern hemisphere mid-latitudes, the best temporal and spatial coverage of the satellite data sets is found and therefore, 27 out of the 33 satellite data sets are considered in this comparison.

### 4.2.1 Antarctic (80°–70°S):

Figure 2 shows the de-seasonalised water vapour time series for the southern polar latitudes. The HIRDLS, SCIAMACHY (solar occultation) and SAGE III observations have no coverage in this latitude region while the GOMOS observations have too limited coverage to allow a derivation of de-seasonalised time series. In the de-seasonalised time series, a spread among the data sets can be found at the four altitudes considered in the comparison. The largest anomalies and the largest spread are





found at 0.1 hPa (up to ±2 ppmv) while the smallest anomalies and thus the smallest spread is found at 3 hPa (generally in the range of ±0.4 ppmv).

At 0.1 hPa the time series start from 1991 onwards with HALOE, since SAGE II measurements are not available at this altitude. Large differences in the seasonal variation of the de-seasonalised time series are found, resulting in a considerable

spread among the data sets, larger than at other altitudes. Large anomalies (up to ±2 ppmv) and thus also a large inter-annual variation are found for the MIPAS-Oxford V5H, MIPAS-ESA V5R and MIPAS-ESA V7R data sets while quite small anomalies are found for both ACE-FTS data sets. These large anomalies in the above mentioned MIPAS data sets are a consequence of the pronounced (spiky) seasonal variation in the absolute data (see Fig. S1 in the Supplement) that is difficult to be accounted for in the sinusoidal regression used for the de-seasonalisation.

Decadal changes in water vapour are found in the de-seasonalised time series at 3 hPa. Several periods of water vapour increases are followed by water vapour decreases. Negative anomalies are found around 1992 while positive anomalies are found around 1996 (HALOE). Water vapour is then showing again positive anomalies in ∼2003 (HALOE, POAM III, SAGE II), followed by a decrease in 2003–2004 which again is followed by a slight increase in water vapour until 2010. From 2010 onwards water vapour remains unchanged. The last increase in water vapour is most strongly pronounced in SMR 489 GHz indicating

a drift in the SMR 489 GHz relative to the other data sets (see also Sect. 4.5). A large spread between the de-seasonalised time series is found between 1999 and 2004 (mainly between POAM III, SAGE II and SMR 489 GHz). Between 2005 and 2014 a good agreement between the de-seasonalised time series is found. However, SMR 489 GHz has somewhat higher anomalies (from 2011 onwards) than the other satellite data sets.

At 10 hPa, the spread among the data sets is quite similar to that observed at 3 hPa, but the variability in water vapour is

more pronounced. There is a decrease in the SAGE II de-seasonalised water vapour time series from 1986–1990. An increase in the de-seasonalised water vapour time series is found in POAM III around 2001. Also from 2009 onwards there seems to be a slight increase in water vapour in all data sets. The SMR 489 GHz de-seasonalised time series at 10 hPa is in good agreement with the de-seasonalised time series of the water vapour products derived from the other satellite instruments. However, the SMR 489 GHz as well as the SOFIE anomalies are low relative to MLS. This becomes quite obvious at the end of the time

series (2012–2014) where only ACE-FTS, MLS, SMR 489 GHz and SOFIE were measuring. Also the influence of the QBO is clearly visible at this altitude level. Distinct positive anomalies are found in 2007–2008 and 2011 and 2013.

At 80 hPa the water vapour distribution is strongly influenced by dehydration (Sect. 4.1). The de-seasonalised time series at 80 hPa once again depict the spread between the individual instruments in this latitude band. At 80 hPa similar results as for 10 hPa are derived (except that here no long-term changes are visible). However, here the deviations between HALOE and

SAGE II are smaller than at 10 and 3 hPa. As at 10 hPa, a decrease in the anomalies of the SAGE II de-seasonalised time series is found from 1986–1990. The de-seasonalised time series then remains constant until 1998 (HALOE and SAGE II). From 1998 onwards the spread between the data sets increases. There is an increase in the anomalies found in 2001 which is followed by a decrease until 2004. Another decrease in water vapour is found in 2009. At 80 hPa, POAM III shows a stronger inter-annual variation and higher/lower anomalies than at 10 and 3 hPa dependent on which year is considered.




### 4.2.2 Tropics (15°S–15°N):

Figure 3 shows the de-seasonalised water vapour time series for the tropics. The POAM III, SAGE III, SCIAMACHY (solar and lunar occultation) and SOFIE data sets have no coverage in this latitude band. In the SAGE II time series some data gaps occur which are due to the aftermath of the Pinatubo eruption (resulting in unrealistically high water vapour values that were filtered out) as well as the so-called "Short Events" between June 1993 and April 1994 where too few measurements were available (Taha et al., 2004). In the tropics, a good consistency between the data sets is found except at 0.1 hPa where again the spread between the data sets is largest. At 0.1 hPa some data sets exhibit larger anomalies ($\pm 1.2$ ppmv as e.g. MIPAS-Oxford V5H and MIPAS-ESA V7R) while others exhibit rather small anomalies ($\pm 0.3$ ppmv as e.g. ACE-FTS and MLS). The HIRDLS, GOMOS and MAESTRO (80 hPa) data sets show generally larger anomalies and thus a larger spread than the other satellite data sets. The de-seasonalised time series in the tropics reflect the decadal changes in water vapour that have been documented in the literature as e.g. the drop in stratospheric water vapour after 2000 and in 2012 (Randel et al., 2004, 2006; Urban et al., 2014). Further, at 3 and 10 hPa, a variability in water vapour on an approximate 2-year timescale associated with the QBO is clearly visible.

At 0.1 hPa the time series starts in 1991 with the HALOE data set that is also the only data set available at this altitude and latitude regions until 2001. The de-seasonalised time series from HALOE shows an increase between 1992–1996 followed by a period with rather constant anomalies until 2001. Afterwards a decrease is visible until 2005. SMR 489 GHz observes, in contrast to HALOE, an increase in water vapour between 2001–2005. Therefore, at the beginning of the SMR 489 GHz record the anomalies at 0.1 hPa are clearly lower as those from HALOE or the other satellite data sets measuring from 2001 onwards. However, a large spread between the data sets is also found during this time period. A similar increase (but somewhat stronger) is found in the MIPAS Oxford V5H data set between 2001–2003, but here the anomalies are higher than the ones from the other satellite data sets. While the MIPAS Oxford V5H and SMR 489 GHz show increasing anomalies, the other data sets show decreasing anomalies. From 2006 onwards all data sets show increasing anomalies. Between 2012–2014, ACE-FTS, MLS and SMR 489 GHz are the only data sets covering this time period and deviations among them are quite visible. SMR 489 GHz anomalies are higher and show a larger inter-annual variability than ACE-FTS and MLS. MLS (together with ACE-FTS) exhibit generally the lowest anomalies ($\pm 0.3$ ppmv) compared to the other satellite data sets at this altitude.

At 3 and 10 hPa the time series begins with SAGE II in 1986. From 1991 onwards HALOE observations are also available. Both, SAGE II and HALOE provide here a much better representation of the temporal development of the water vapour time series and the inter-annual variability than in the Antarctic since both data sets have a much better temporal coverage in the tropics (see Figs. S1 and S2 in the Supplement). SAGE II shows somewhat larger anomalies than HALOE. Generally, the de-seasonalised time series show a good agreement with each other at these two altitude levels (3 and 10 hPa). Further, at these altitude levels, the lowest anomalies and the lowest spread between the data sets is found, especially at 10 hPa. The deviations between MLS (or ACE-FTS) and SMR 489 GHz found during the time period 2012–2014 are still evident at 3 hPa but to a much lesser extent than at 0.1 hPa. At 3 hPa, inter-annual variations (with anomalies roughly in the order of $\pm 1$ ppmv) due to the QBO are clearly visible. At 10 hPa this variability is far less obvious. Also, the differences between SMR 489 GHz



and the other data sets measuring during the time period 2001–2005 (SAGE II and HALOE) are found to a lesser extent at 3 hPa, but not at 10 hPa. The GOMOS data set exhibits large scatter. At 10 hPa the HIRDLS data set indicates stronger inter-annual variability than the other satellite instruments. This level is the uppermost altitude where HIRDLS can be retrieved and accordingly the data here are more uncertain. Both drops in water vapour, the one in 2001 and the one in 2012 are clearly

visible in the de-seasonalised time series at 10 hPa. The latter one is strongly pronounced in the three remaining data sets covering that time period (ACE-FTS v3.5, MLS and SMR 489 GHz). There is also a clear variability on an approximate 2-year timescale associated with the QBO visible at this altitude level, however not at all times as clearly pronounced as at 3 hPa.

Similar to the other three pressure levels, at 80 hPa relatively good agreement between SAGE II and HALOE is found. However, SAGE II typically shows somewhat lower anomalies than HALOE. At 80 hPa, a higher variability with larger

anomalies than at 10 and 3 hPa are found (generally around ± 0.8 ppmv). The data sets agree well in terms of the inter-annual variation. The drops in 2000 and 2011 are consistently observed, as are the recoveries afterwards. This is also true for the pronounced QBO in 2006–2008. In 2005 the MIPAS-Bologna V5R NOM and MIPAS-ESA V5R NOM data sets show strong negative anomalies (up to -2 ppmv) which are not found in the other data sets. A similar behaviour of these data sets is found in 2011, where these data sets show strong positive anomalies (up to 1.6 ppmv) while in the other satellite data sets only

anomalies up to 0.4–0.8 ppmv are found. MAESTRO shows strong scatter, mainly because 80 hPa is near the upper altitude limit of the MAESTRO water vapour retrieval. Another distinctive characteristic in the de-seasonalised time series at 80 hPa is the increase in water vapour until mid 2014 (ACE-FTS v3.5, MLS and SMR 544 GHz) which is anti-correlated to the time series at 10 hPa.

### 4.2.3 Northern mid-latitudes (50°–60°N):

Figure 4 shows the de-seasonalised time series for the northern mid-latitudes. The GOMOS, SCIAMACHY lunar and SOFIE data sets have no coverage in this latitude region. As for the other latitude bands the largest spread between the satellite data sets is found at 0.1 hPa. This is accompanied by a large inter-annual variability. The ACE-FTS v3.5, MIPAS-Bologna V5H, MIPAS-Oxford V5H and SMR 489 GHz data sets are among the data sets showing the largest inter-annual variability and also the largest anomalies at 0.1 hPa. The MIPAS-Oxford V5H data set covers the time period from 2002–2004 and here the largest

anomalies (exceeding 2 ppmv) are found. The largest negative anomalies are found in 2005 and 2006 with −1.6 and −2 ppmv, respectively. The differences between ACE-FTS v3.5 and the other satellite data sets become most pronounced at the end of the data record when only SMR 489 GHz and MLS were still measuring. Here, ACE-FTS v3.5 shows some larger variability. At this altitude, the drift in the SMR 489 GHz data set is again visible. Until 2004 the anomalies are typically more negative compared to the other data sets, while they are more positive after 2012. The HALOE data set indicates an increase in water

vapour until about 1997 and a decrease afterwards. From 2007 to 2010 there appears to be decrease in water vapour for all data sets while there is a pronounced increase after that until early 2012.

At 3 hPa, the de-seasonalised time series show generally good agreement while at 10 hPa the best agreement is found. Differences at 3 hPa are that SMR 489 GHz exhibits lower anomalies during the time period 2001 to 2006 and higher anomalies than the other data sets from 2010 to 2014 and that SAGE II shows higher anomalies than the other satellite instruments at





the end of their data record (2004–2005). Differences at $10\,hPa$ are found in the time period 2004–2008 where SAGE II and HIRDLS show a stronger inter-annual variability and between 2010–2012 where SMR $489\,GHz$ exhibits somewhat higher anomalies than the other satellite data sets. In both altitude levels, an increase in water vapour between 1992 and 2000 ($10\,hPa$) and 1992 and 1998 ($3\,hPa$), respectively, is found. The two water vapour drops that occurred after 2000 and in 2011 in the

tropics (Randel et al., 2004, 2006; Urban et al., 2014) are also visible at $10\,hPa$ in the northern hemisphere mid-latitudes, however with a temporal delay.

Although the inter-annual and decadal variability at $80\,hPa$ is low, some satellite data sets (MAESTRO, POAM III and SMR $544\,GHz$) show larger deviations from the other satellite data sets. In the MAESTRO data, a high inter-annual variability is found with anomalies reaching up to $1.6\,ppmv$. In this altitude regions, MAESTRO has its best temporal coverage in the mid-

latitudes, but still $80\,hPa$ is at the upper limit of the MAESTRO measurements and therefore not every measured profile reaches that high up. This explains why a higher variability (scatter) than in the other satellite data sets is found for the MAESTRO time series. POAM III exhibits much larger anomalies than the other satellite data sets ($+1.2\,ppmv$ compared to $\pm0.4\,ppmv$). Although the POAM III anomalies decrease with time, they still remain higher than the anomalies from the other satellite data sets. The differences between POAM III and the other satellite data sets are caused by the limited temporal sampling (only

summer months are measured) of POAM III in this latitude region making the de-seasonalisation by regression apparently fail. In the SMR $544\,GHz$ data set, a larger inter-annual variability is found, but with much smaller anomalies than MAESTRO. In the SAGE II data, the anomalies are slightly decreasing in the time period 1987–2002. Further, there is some pronounced QBO variation alongside an overall increase from 2004 to 2012.

Overall, in the northern hemisphere mid-latitudes, the lowest inter-annual variability is found, especially at $80\,hPa$. Similar

to the comparisons in the Antarctic and tropics, the largest inter-annual and decadal variability as well as the largest spread between the data sets is found at $0.1\,hPa$. The drops in stratospheric water vapour after 2000 and in 2011 (Randel et al., 2004, 2006; Urban et al., 2014) that are observed in the tropics are also found at $10\,hPa$ in the mid-latitudes, but with a temporal delay and to a lesser extent than in the tropics.

### 4.3  Spread assessment

In the following, the spread between the data sets is quantitatively assessed to provide an estimate of the uncertainty in the observational database. Fig. 5 shows the difference between the maximum and minimum volume mixing ratio among the different de-seasonalised water vapour data sets as a function of time and altitude for the three latitude bands, Antarctic, tropics and northern hemisphere mid-latitudes. The spread of the absolute time series is shown in the Supplement in Fig. S7. The spread is calculated for the years 2000–2014. Earlier years are not considered due to the lack of a sufficient number of satellite

instruments measuring during that time period. Before 2000 only HALOE, POAM III and SAGE II data were available which results in a too sparse and not meaningful picture (similar to the gaps found for the early years in Fig. 5). The spread estimates become more meaningful as more satellite data sets are available. This can be seen from Fig. 5 for the years from 2002 onwards. For the years 2000–2001 and 2012–2014 between two and four data sets were available. In these cases the differences among



the data sets are not as pronounced and probably less meaningful than for the years 2002–2012 where the majority of satellite instruments were measuring.

In all three latitude bands the spread is large at the highest and lowest altitude level considered in this study which correspond to the upper troposphere/tropopause region and the lower mesosphere. The large spread in these altitude regions is on one hand

related to large uncertainties in the water vapour observations (e.g. due to increased measurement noise) and on the other hand also to the variability of the atmosphere and its different representation in the individual data sets. In addition, a large spread is found in the Antarctic lower stratosphere (Fig. 5 top) in winter and spring where the water vapour distribution in the lower stratosphere is affected by dehydration and transport of low water vapour from the mesosphere into the stratosphere (Section 4.1). In the tropics (Fig. 5 middle), the lowest spread compared to the other latitude bands is found. Increased values

are found here as in the other regions at the highest and lowest levels. The spread is lowest in the time period 2006 to 2010. A similar behaviour is found for the mid-latitudes (Fig. 5 bottom), also here the spread seems to be lower around 10 hPa during the time period 2006–2010. The mid-latitudes show features similar to the tropics and polar regions. In the northern hemisphere mid-latitudes, the largest spread occurs in the lower stratosphere where low water vapour is found due to air masses that are freeze dried when entering the stratosphere in the tropics (atmospheric tape recorder), and in the lower mesosphere due to the

descent of air within the polar vortex.

### 4.4 Correlation assessment

To assess the temporal consistency between individual data sets, the correlation coefficient between all possible combinations of data sets is considered. In this section, the results for the de-seasonalised time series are presented while the results for the absolute time series are given in the Supplement. We start by presenting an example correlation of the MIPAS-Oxford V5R NOM

time series with those from the other data sets and then present all correlations in form of matrices.

### 4.4.1 Correlation example

Figure 6 shows the correlation between the de-seasonalised MIPAS-Oxford V5R NOM time series and those from the other data sets for the Antarctic, tropics and the northern hemisphere mid-latitudes. The largest spread in the correlation between the satellite data sets is found in the Antarctic (Fig. 6 top). Here, also the lowest correlation over all altitude levels is found

(rarely exceeding a correlation coefficient of 0.8). MIPAS-ESA V5R NOM and MIPAS-ESA V7R are among the data sets showing the highest correlation with MIPAS-Oxford V5R NOM over all altitude levels while the lowest correlation with MIPAS-Oxford V5R NOM is found for SCIAMACHY lunar throughout most altitudes. The SOFIE and SMR 544 GHz data sets show very low correlations (even negative for SOFIE) at the lowest altitude levels (below 10 hPa) as well as above 3 hPa (but here SMR 489 GHz instead of SMR 544 GHz). In between these altitudes levels the SOFIE and SMR 489 GHz data sets

show a similar correlation to MIPAS-Oxford V5R NOM than the other data sets.

In the tropics (Fig. 6 middle), the correlation coefficients vary between 0.8 to 1 for most data sets in the altitude region (30 to 1 hPa). Low correlations are found for all data sets between 100 hPa and 30 hPa, except the MIPAS-IMKIAA V5R NOM data set that shows a high correlation (>0.8) up to 1 hPa with MIPAS-Oxford V5R NOM. The data sets that show the lowest





correlation with MIPAS-Oxford V5R NOM (even in some occasions negative) are GOMOS and MAESTRO. These data sets thus deviate from the typical correlation of most other data sets. Above 60 hPa and above 25 hPa this is also true for HIRDLS and SMR 544 GHz, respectively. These two data sets show at the lowest altitude levels a reasonable correlation with MIPAS-Oxford V5R NOM, but then the correlation coefficients decrease rapidly with increasing altitude, most likely

due to increased measurement noise. At altitudes above 0.7 hPa the correlation decreases for all data sets and also the spread between the data sets increases. For MIPAS-ESA V5R NOM, the correlation, although decreasing, remains rather high with a correlation coefficient of 0.7. The lowest correlation at 0.1 hPa is found for the ACE-FTS v2.2, ACE-FTS v3.5, MIPAS Bologna V5R NOM and MIPAS-Bologna V5R MA data sets.

   In the northern hemisphere mid-latitudes (Fig. 6 bottom), the correlation coefficients varies between 0.4 and almost 1 in the

altitude region between 0.7 hPa and 10 hPa depending on which data set is considered. The spread in the northern hemisphere mid-latitudes is almost as large as the spread in the Antarctic. A very high correlation (correlation coefficient of around 0.9–1) between MIPAS-Oxford V5R NOM and the other data sets is found e.g. at around 1 hPa for the MIPAS-ESA V5R NOM and MIPAS-ESA V7R data sets. The lowest correlation between MIPAS-Oxford V5R NOM and the other data sets is found above 1 hPa for the two ACE-FTS data sets while the SMR 489 GHz data set shows a rather low correlation throughout the

entire altitude region considered in this study. Below 10 hPa the lowest correlations (even negative correlations) are found for HIRDLS, MAESTRO, SCIAMACHY limb and SMR 544 GHz data sets. These data sets also deviate from the usual spread in correlation of the data sets.

### 4.4.2   Correlation matrices

The correlation of all data sets is given in Fig. 7–9 in form of matrix plots for the three latitude bands and four altitude levels. In

addition to the correlation coefficient, the number of months the time series actually overlap is also given (requiring a minimum of 12 months, see Sect. 3.2.2). The same figures for the correlation of the absolute time series are given in the Supplement (Fig. S8–S10). The correlation matrix shown in Fig. 7 gives a good overview over the temporal consistency of all data sets in the Antarctic. The correlations between the data sets are generally positive (green), but in some cases negative correlations (red) are found, this is e.g. the case for the correlation between the MIPAS-IMKIAA V5H and POAM III data sets at 10 hPa

or between the MLS and SCIAMACHY lunar data sets at 3 hPa. However, in these two cases, the number of months the data sets overlapped were not that high (14 and 28) and this may explain the low correlation between these data sets. An example where despite a high number of overlapping months (70) a negative correlation is found is the correlation between the MIPAS-Bologna V5R NOM and MLS data sets at 0.1 hPa. An example for a high number of overlapping months (114) and high correlation coefficient is the correlation between the MLS and SMR 489 GHz data sets at 10 hPa. Nevertheless, although in

the Antarctic the correlation is generally positive, the correlation coefficient rarely exceeds 0.5. An exception is the 3 hPa level where a generally high correlation among the MIPAS data sets is found. A similar behaviour between the MIPAS data sets is found at 10 hPa.

   In Fig. 8 the correlation matrix for the tropics is shown. The large spread between the data sets we found in Fig. 6 at 0.1 hPa is also reflected in the correlations among all data sets. The same holds for the good correlations that are found at 3



and 10 hPa. An exception here is the GOMOS data set that shows negative correlations with all instruments at 3 hPa, but the number of months the data sets actually overlapped is rather low. At 80 hPa the spread between the data sets is not as large as at 0.1 hPa, but still larger than at 3 and 10 hPa. At 80 hPa occasionally negative correlations are found. This primarily concerns comparisons involving the GOMOS, HALOE, MAESTRO and MIPAS-Oxford V5H data sets. The lowest (negative)

correlation is found between SMR 489 GHz and SAGE II data sets, but also here the number of months the data sets had overlap was rather low with 21 months.

The correlation matrix shown in Fig. 9 gives a good overview over the temporal consistency of all data sets in the mid-latitudes. The majority of the correlations are positive, but for some comparisons a negative correlation is found. One such example is the correlation between the MIPAS Bologna V5H and SMR 489 GHz data sets at 3 hPa. However, again the num-

ber of months the data sets actually had overlap was rather low and may explain the negative correlation between these data sets. An example where, despite a high number of overlapping months, a negative correlation is found is the correlation between MIPAS-Bologna V5R NOM and MIPAS-Bologna-V5R-MA with MLS at 0.1 hPa. The correlation of these two data sets with the other data sets is also generally low at 0.1 hPa. Also for the two ACE-FTS data sets the correlation of most data sets is often low despite a sufficient number of overlapping months. Positive correlations are found for the ACE-FTS v2.2/v3.5

data sets in comparison to the MIPAS-IMKIAA V5R MA, MIPAS-Oxford V5R MA, MLS and SMR 489 GHz. The highest correlation at 0.1 hPa is found between the two ACE-FTS data sets and between ACE-FTS v2.2 and MLS. At 3 and 10 hPa generally a high correlation among the MIPAS data sets is found. At 10 hPa the correlation of HIRDLS with some data sets is high, but low with the other data sets. At 80 hPa low correlations between MAESTRO and all other instruments are found.

In summary, a high number of overlapping months does not necessarily guarantee a good correlation between two data sets, but generally the chances are quite high if this is the case. On the other hand, if data sets overlap only a low number of months still a good agreement between these data sets can be found. Therefore, for assessing the agreement between two data sets both quantities should be taken into account. The correlation assessment again confirms what we found before from the qualitative time series comparison, namely that the best agreement between the satellite data sets is found in the tropics while in Antarctic

and northern hemisphere mid-latitudes a large spread between the data sets is found. Generally, the lowest correlations are found in the Antarctic. Further, in each latitude band the correlation is lower in the lower stratosphere and lower mesosphere than in the middle stratosphere.

## 4.5   Drift assessment

In addition to the spread and correlations, the drifts among the satellite data sets are considered. As drift we consider the

linear change of the difference between two time series, which indicates if the longer-term variation of the two time series is the same or not (Sect. 3.3). As before we start with an example. In Fig. 10 the drifts between the de-seasonalised time series of the SMR 489 GHz and all other data sets are shown for the northern hemisphere mid-latitudes (left panel) as well as the corresponding significance level (right panel). The significance level is given by the absolute ratio of the drift to the drift





uncertainty. We consider a drift as statistically significant when the significance level is larger than $2\sigma$ (corresponding to the 95% confidence level).

### 4.5.1 Drift example

Figure 10 shows that below 20 hPa large drifts (up to $2.5\,\mathrm{ppmv\,decade^{-1}}$ and even higher) are found between SMR 489 GHz and the other satellite data sets. In the altitude region between 20 hPa and 1 hPa, a good consistency between the satellite data sets is found despite the different time periods of measurements. Around 20 hPa the smallest drifts are found, ranging from about 0 to $0.5\,\mathrm{ppmv\,decade^{-1}}$. The drifts are consistently increasing with altitude and maximise around 0.4 hPa. Above 1 hPa the drifts of SMR 489 GHz vary between about 0.75 and $1.5\,\mathrm{ppmv\,decade^{-1}}$ depending on which data set the SMR 489 GHz data set is compared to, but decrease with altitude towards 0.1 hPa. The drifts range here between 0 and $1.25\,\mathrm{ppmv\,decade^{-1}}$. The drifts between SMR 489 GHz and the other satellite data sets is in most cases significant at the $2\sigma$ uncertainty level as can be seen from Fig. 10 (right panel). Larger drifts between SMR 489 GHz and the other data sets that obviously deviate from the majority of data sets are found for the comparison to the POAM III, SAGE II, SAGE III and HALOE data sets. However, this is due to the fact that for these data sets not only the overlap period with SMR 489 GHz is relatively short (4 years, from 2001–2005), but also the number of months were both data sets actually yield a valid monthly mean is small (see numbers given in figure legend). Additionally, these drifts are in most cases not statistically significant at the $2\sigma$ uncertainty level.

### 4.5.2 Drift matrices

In Fig. 11–13 the drift estimates between the time series of all data sets are summarised as matrix plots for the three latitude bands and four altitudes. In the matrix plots, data sets are only shown if they yield any result at a given altitude. The drift estimates are based on the difference time series between the data sets given at the x-axis and the data sets given at the y-axis. Additional information that is given in the matrix plots includes the overlap period of the two data sets, how many months the data sets actually overlap and if the drift is significant or not at the $2\sigma$ uncertainty level as well as the corresponding significance level for a significant drift.

In the Antarctic (Fig. 11), almost no significant drifts are found between the satellite data sets at the two lowest altitude levels (80 and 10 hPa). An exception here is the MAESTRO data set which shows a significant (negative) drift of $-2$ to $-3\,\mathrm{ppmv\,decade^{-1}}$ (significance level up to 3.7) and POAM III which shows a significant positive drift ($2$ to $3\,\mathrm{ppmv\,decade^{-1}}$) compared to SAGE II and SMR 544 GHz (at 80 hPa). While the overall time period MAESTRO overlapped with other data sets was sufficiently long (>85 months), the number of coincident months for these data sets was rather low (9 months). Further, at 80 hPa, a significant negative drift is found between some MIPAS data sets and SOFIE. At 10 hPa, a significant (positive) drift ($0.8\,\mathrm{ppmv\,decade^{-1}}$) is found between the MIPAS-Oxford V5R NOM and ACE-FTS v2.2 data sets (significance level of 3.2) and of $2\,\mathrm{ppmv\,decade^{-1}}$ between the SMR 489 GHz and POAM III data sets (significance level 3.0). Additionally, significant drifts are found between different MIPAS data sets relative to SMR 489 GHz and between the MLS and SMR data sets. At 3 hPa most drifts are significant. Most MIPAS data sets exhibit significant positive drifts relative to the ACE-FTS (significance level up to 5.7) and MLS (significance level up to 8.1) data sets. While in the comparisons to the ACE-FTS data sets the actual





number of overlapping months is limited, this is not the case in the comparison to MLS. As before, for the SMR 489 GHz data set significant positive drifts are found (significance level up to 4.8) relative to most other data sets. A large variety of drifts is found at 0.1 hPa, but in most cases the drift is not significant. Data sets for which most drifts are significant at this altitude level are SMR 489 GHz (>2 ppmv decade$^{-1}$, significance level up to 6.4) and MIPAS-Bologna V5R MA (significance level

up to 3.2).

In the tropics (Fig. 12), larger drifts are found than in the Antarctic, especially at 0.1 hPa. Here, most drifts are significant. Significant drifts are found for the MIPAS-Bologna V5R NOM, MIPAS-Bologna V5R MA, MIPAS-ESA V5R, MIPAS-IMKIAA V5R NOM, MIPAS-Oxford V5R NOM and SMR 489 GHz data sets. For example, for MIPAS-Bologna V5R NOM and MIPAS-Bologna V5R MA a drift (significance level up to 6.5) in comparison to most other satellite data sets is found.

For MIPAS-Bologna V5R NOM this is also the case at 3 hPa (significance level up to 9.8). Large negative drifts are found for GOMOS (> −2.5 ppmv decade$^{-1}$, significance level up to 3.9) compared to most data sets. Also for SMR 489 GHz significant positive drifts (up to ∼1 ppmv decade$^{-1}$, significance level up to 8.5) to almost all data sets are found at 3 hPa. A good consistency is found among the MIPAS data sets. The drifts are low and in most cases not significant. An exception here is MIPAS-Oxford V5R NOM (∼0.6–1 ppmv decade$^{-1}$, significance level up to 9.8). For the tropics the best agreement among

the data sets is found at 10 hPa. In most cases the drift is not significant and in cases where the drift is significant the drifts are relatively low with 0.2–0.4 ppmv decade$^{-1}$. Larger drifts are found at this altitude for GOMOS (up to −3 ppmv decade$^{-1}$) and HIRDLS (up to −2 ppmv decade$^{-1}$). For GOMOS the drifts are most cases significant (significance level up to 4.3) while this is not the case for HIRDLS.

At 80 hPa a wide variety is found. Some data sets show a positive drift, some a negative. In some cases the drift is sig-

nificant and in other cases not. For example, a positive drift (2 ppmv decade$^{-1}$) relative to almost all data sets is found for MIPAS-Bologna V5R NOM (significance level up to 6.4). For the HIRDLS data set a significant positive drift (also ∼2 ppmv decade$^{-1}$) is found compared to MIPAS-IMKIAA V5R NOM, MIPAS-IMKIAA-V5R MA and MIPAS-Oxford V5R NOM (significance level 2.0–4.6). A large drift (>3 ppmv decade$^{-1}$) at this altitude level is found for MIPAS-ESA V5R MA compared to MIPAS-IMKIAA V5R NOM (significance level 4.8). Also the MIPAS-Oxford V5R NOM shows significant drifts

compared to a number of data sets.

The pattern of the estimated drifts in the northern hemisphere mid-latitudes shown in Fig. 13 are quite similar to the drifts in the tropics and Antarctica. However, the estimated change in ppmv decade$^{-1}$ seems to be somewhat lower in the mid-latitudes than in the tropics or Antarctic. The highest variety is again found at 0.1 hPa. Similar to the tropics significant drifts are found for e.g. the MIPAS-Bologna V5R NOM and MIPAS-Bologna V5R MA (up to −2 ppmv decade$^{-1}$, significance level up to

3.9) data sets relative to the SMR 489 GHz data set. At 3 hPa, for most data sets the drifts are small and/or not significant. Significant negative drifts are found for both ACE-FTS data sets and for SMR 489 GHz. For SMR 489 GHz a drift is found relative to most other data sets which is also in most cases significant. At 10 hPa HIRDLS shows pronounced drifts compared to the other data sets. However, these drifts are not significant except for the comparison with MLS (drift of 3 ppmv decade$^{-1}$, significance level 2.3). Otherwise for most data sets the drifts are small and/or not significant at 10 and 80 hPa. An exception

are HIRDLS (−2 ppmv decade$^{-1}$) and MAESTRO (−1 ppmv decade$^{-1}$) which show a negative drift at 80 hPa. For HIRDLS



in most cases the drift is significant (significance level up to 4.1), but for MAESTRO in most cases not. For MIPAS-Bologna-V5R NOM significant positive drifts are found to all instruments which are in the most cases around 0.2–0.4 ppmv decade$^{-1}$, but higher compared to HIRDLS (significance level 4.1), MAESTRO (significance level 2.2), SCIAMACHY limb (significance level 10.6) and SCIAMACHY solar OEM (significance level 6.6). Other data sets for which drifts are found compared to most

other data sets are SCIAMACHY limb, SCIAMACHY solar Onion and SMR 489 GHz.

## 5  Summary and Conclusions

In the framework of the second SPARC water vapour assessment, time series of stratospheric and lower mesospheric water vapour derived from satellite observations were compared. The comparison results presented comprise 33 data sets from 15 satellite instruments. These comparisons provide a comprehensive overview of the typical uncertainties in the observational

database which should be considered in the future in observational and modelling studies addressing stratospheric and lower mesospheric water vapour variability and trends.

The time series comparison was performed for three latitude bands: the Antarctic (80–70°S), the tropics (15°S–15°N) and the northern hemisphere mid-latitudes (50°–60°N) at four altitudes levels (0.1, 3, 10, 80 hPa) covering the stratosphere and lower mesosphere. The combined temporal coverage of observations from the 15 satellite instruments allows considering the

time period 1986–2014. In addition to the qualitative comparison of the time series, a quantitative comparison was provided based on the spread, correlation and drift between the individual time series.

The qualitative time series comparison shows that the largest differences between the de-seasonalised time series are in the Antarctic and in the lower mesosphere (0.1 hPa) and tropopause region (80 hPa). In the stratosphere (3 and 10 hPa) and the tropics, good agreement between the satellite data sets was found. These differences were quantitatively confirmed by the

correlation assessment where the best agreement between the satellite data sets was also found in the tropics while in Antarctic and northern hemisphere mid-latitudes a large spread between the data sets was found. Generally, the lowest correlations between the individual data sets was found in the Antarctic. In each latitude band the correlation was lower in the lower stratosphere and lower mesosphere than in the middle stratosphere.

The reason why the largest differences between the data sets are found in the tropopause region and the lower mesosphere as

well as in the Antarctic is because here also the highest variability in water vapour is found. Given the limited vertical resolution of the satellite data sets, tropospheric influences start to play a role near the tropopause. Sampling differences become more pronounced due to the large variability, e.g. due to the fact that the satellite observations are differently influenced by clouds. In the lower mesosphere, diurnal variation becomes more important. The satellite data sets do not have the same local time coverage. For example there is an influence of non local thermodynamic equilibrium effects (NLTE) in most MIPAS data sets

except MIPAS-IMKIAA V5R MA where these NLTE effect are explicitly considered. Another example for larger deviations in the lower mesosphere are the MIPAS NOM data sets that are at this altitudes close to their upper retrieval limit and thus more uncertain.



Less agreement between the data sets was found for the Antarctic, especially in the lower stratosphere in winter and spring when dehydration occurs. Large differences between the data sets were found in both, the absolute and de-seasonalised data. In the absolute data, these differences are primarily caused by differences in the influence of clouds on the measurements. However, sampling biases can also play a role. In the de-seasonalised data some differences between the data sets can be

related to the approach for the de-seasonalisation used in our study (e.g. POAM III). Since the dehydration is more a seasonal phenomenon, the regression with sinusoidal functions is problematic. For these data sets the average approach (see Sect. 3.1) would be the more adequate approach for de-seasonalisation.

In addition to the assessment of the spread and correlations, the drifts between the individual data sets were also assessed which indicates if the longer-term variations (drifts) of two time series are the same or not. From the drift comparison we

found that the drift patterns are quite similar for the three latitude bands considered. The drifts are highest at the highest and lowest considered altitude level ($0.1\,\mathrm{hPa}$ and $80\,\mathrm{hPa}$). Most significant drifts were found in the tropics which coincides with low spread/variability which makes drift detection considerably easier. Further, it is possible that some of the drifts (especially for the low-density samplers) are caused by sampling biases (Damadeo et al., 2018). The same drifts as shown here were also calculated from profile-to profile comparisons (using coincident data) by Lossow et al. (in preparation). However, no

statistically significant difference was found between the two sets of drifts in 95% of the comparisons.

Further, from the drift assessment we found that the MIPAS data sets show positive drifts relative to the ACE-FTS data sets in the Antarctic and northern mid-latitudes at $3\,\mathrm{hPa}$. Interestingly, no drifts of MIPAS relative to ACE-FTS are found in the tropics. The reason for this is currently not understood. The drifts found in the MIPAS data sets are consistent with the unaccounted time dependence of the correction coefficients for the non-linearity in the detector response function used in the data sets based

on calibration version 5 (Walker and Stiller, in preparation). Some improvement is seen in the MIPAS ESA V7R NOM data set where a time dependence of the correction coefficient is implemented, however, not at all altitudes. Additionally, even drifts among the different MIPAS data sets were found. This might be related amongst others to the different retrieval choices (as well as to the usage of different micro-windows) by the different processors and to sampling differences between the NOM and MA observations. Further, from the drift comparison, we found that SMR $489\,\mathrm{GHz}$ data sets has a significant drift relative to

the other data sets, except at around $10\,\mathrm{hPa}$. The drifts of the SMR $489\,\mathrm{GHz}$ data set are largest at around $50\,\mathrm{hPa}$ and $0.5\,\mathrm{hPa}$ with approximately 1.5 and $>2\,\mathrm{ppmv\,decade^{-1}}$, respectively, dependent on the data set used for comparison.

Further, within this assessment study we encountered the following difficulties in our analyses using the HIRDLS, GOMOS and MAESTRO data sets. The GOMOS time series exhibit larger scatter from month to month (coverage only in the tropics for de-seasonalised data here), despite extended screening (Walker and Stiller, in preparation) resulting in low correlations

to the other data sets and pronounced negative drifts at $10\,\mathrm{hPa}$ and $3\,\mathrm{hPa}$. The quality of the HIRDLS data set deteriorates towards $10\,\mathrm{hPa}$ resulting in low correlations and larger anomalies as well as larger drifts. However, the drifts mostly were not statistically significant. It should be noted here that additionally to correcting for the effects of the obstruction in the optics, changes in the calibration were made in between the HIRDLS mission (Gille et al., 2008, 2012). This change in calibration may also have an influence on the drift estimates. The MAESTRO data set exhibits large uncertainty (noise) at $80\,\mathrm{hPa}$ (in the





correlations and drifts) which is related to the vicinity to the uppermost limit of these retrievals. A similar behaviour is also found for the SCIAMACHY limb and the SMR 544 GHz data sets.

Nevertheless, from our assessment we find that all data sets can be considered in the future in observational and modelling studies addressing e.g stratospheric and lower mesospheric water vapour variability and trends when data set specific
characteristics (as e.g. a drift of the instrument) and restrictions (as e.g. spatial and temporal coverage) are taken into account.

**Dedication to Jo Urban**

We would like to dedicate this paper to our highly valued colleague Jo Urban who would have definitely been the lead author of this study if he would not have passed away so early. Without his devoted work on UTLS water vapour over many years this work would not have been possible. In particular, the retrieval of water vapour from the SMR observations and the combination
of these data with other data sets to understand the long-term development of this trace constituent comprised a large part his life's work. With his death, we lost not only a treasured colleague and friend, but also a leading expert in the microwave and sub-millimetre observation community.

**The Supplement related to this article is available online at doi:10.5194/amt-0-1-2018-supplement.**

*Acknowledgements.* The Atmospheric Chemistry Experiment (ACE), also known as SCISAT, is a Canadian-led mission mainly supported
by the Canadian Space Agency and the Natural Sciences and Engineering Research Council of Canada. We would like to thank the European Space Agency (ESA) for making the MIPAS level-1b data set available. MLS data were obtained from the NASA Goddard Earth Sciences and Information Center. Work at the Jet Propulsion Laboratory, California Institute of Technology, was done under contract with the National Aeronautics and Space Administration. SCIAMACHY spectral data have been provided by ESA. The work on the SCIAMACHY water vapour data products has been funded by DLR (German Aerospace Center) and the University of Bremen. The SCIAMACHY limb water
vapour data set v3.01 is a result of the DFG (German Research Council) Research Unit "Stratospheric Change and its Role for Climate Prediction" (SHARP) and the ESA SPIN (ESA SPARC Initiative) project and were partly calculated using resources of the German HLRN (High-Performance Computer Center North). We would like to thank M. Hervig for providing the SOFIE data. We acknowledge the HALOE science team and the many members of the HALOE project for producing and characterising the high quality HALOE data set. Further, we would like to thank E. Remsberg for valuable comments on the manuscript. Stefan Lossow was funded by the SHARP project under contract
STI 210/9-2. We want to express our gratitude to SPARC and WCRP (World Climate Research Programme) for their guidance, sponsorship and support of the WAVAS-II programme. We acknowledge support by Deutsche Forschungsgemeinschaft and Open Access Publishing Fund of Karlsruhe Institute of Technology.



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

©c Author(s) 2018. CC BY 4.0 License.





Remsberg, E.: Observed seasonal to decadal scale responses in mesospheric water vapor, Journal of Geophysical Research, 115, D06 306, doi:10.1029/2009JD012904, 2010.

Remsberg, E., Russell, J. M., Gordley, L. L., Gille, J. C., and Bailey, P. L.: Implications of the stratospheric water vapor distribution as determined from the Nimbus 7 LIMS experiment, Journal of the Atmospheric Sciences, 41, 2934 – 2948, 1984.

Riese, M., Ploeger, F., Rap, A., Vogel, B., Konopka, P., Dameris, M., and Forster, P.: Impact of uncertainties in atmospheric mixing on simulated UTLS composition and related radiative effects, Journal of Geophysical Research, 117, D16 305, doi:10.1029/2012JD017751, 2012.

Rosenlof, K. H., Chiou, E.-W., Chu, W. P., Johnson, D. G., Kelly, K. K., Michelsen, H. A., Nedoluha, G. E., Remsberg, E. E., Toon, G. C., and McCormick, M. P.: Stratospheric water vapor increases over the past half-century, Geophysical Research Letters, 28, 1195 – 1198,
doi:10.1029/2000GL012502, 2001.

Scherer, M., Vömel, H., Fueglistaler, S., Oltmans, S. J., and Staehelin, J.: Trends and variability of midlatitude stratospheric water vapour deduced from the re-evaluated Boulder balloon series and HALOE, Atmospheric Chemistry & Physics, 8, 1391 – 1402, doi:10.5194/acp-8-1391-2008, 2008.

Schiller, C., Grooß, J.-U., Konopka, P., Plöger, F., Silva Dos Santos, F. H., and Spelten, N.: Hydration and dehydration at the tropical
tropopause, Atmospheric Chemistry & Physics, 9, 9647 – 9660, doi:10.5194/acp-9-9647-2009, 2009.

Schoeberl, M. R., Douglass, A. R., Newman, P. A., Lait, L. R., Lary, D., Waters, J., Livesey, N., Froidevaux, L., Lambert, A., Read, W., Filipiak, M. J., and Pumphrey, H. C.: QBO and annual cycle variations in tropical lower stratosphere trace gases from HALOE and Aura MLS observations, Journal of Geophysical Research, 113, D05 301, doi:10.1029/2007JD008678, 2008.

Seele, C. and Hartogh, P.: Water vapor of the polar middle atmosphere: Annual variation and summer mesosphere conditions as observed by
ground-based microwave spectroscopy, Geophysical Research Letters, 26, 1517 – 1520, doi:10.1029/1999GL900315, 1999.

Seinfeld, J. H. and Pandis, S. N.: Atmospheric chemistry and physics: From air pollution to climate change, John Wiley and Sons, New York, second edition, 2006.

Solomon, S.: Stratospheric ozone depletion: A review of concepts and history, Reviews of Geophysics, 37, 275 – 316, doi:10.1029/1999RG900008, 1999.

Solomon, S., Rosenlof, K. H., Portmann, R. W., Daniel, J. S., Davis, S. M., Sanford, T. J., and Plattner, G.: Contributions of stratospheric water vapor to decadal changes in the rate of global warming, Science, 327, 1219 – 1223, doi:10.1126/science.1182488, 2010.

Stiller, G. P., von Clarmann, T., Haenel, F., Funke, B., Glatthor, N., Grabowski, U., Kellmann, S., Kiefer, M., Linden, A., Lossow, S., and López-Puertas, M.: Observed temporal evolution of global mean age of stratospheric air for the 2002 to 2010 period, Atmospheric Chemistry & Physics, 12, 3311 – 3331, doi:10.5194/acp-12-3311-2012, 2012.

Taha, G., Thomason, L. W., and Burton, S. P.: Comparison of Stratospheric Aerosol and Gas Experiment (SAGE) II version 6.2 water vapor with balloon-borne and space-based instruments, Journal of Geophysical Research, 109, D18 313, doi:10.1029/2004JD004859, 2004.

Thölix, L., Backman, L., Kivi, R., and Karpechko, A. Y.: Variability of water vapour in the Arctic stratosphere, Atmospheric Chemistry & Physics, 16, 4307 – 4321, doi:10.5194/acp-16-4307-2016, 2016.

Tweedy, O. V., Kramarova, N. A., Strahan, S. E., Newman, P. A., Coy, L., Randel, W. J., Park, M., Waugh, D. W., and Frith, S. M.: Response
of trace gases to the disrupted 2015-2016 quasi-biennial oscillation, Atmospheric Chemistry & Physics, 17, 6813 – 6823, doi:10.5194/acp-17-6813-2017, 2017.





Urban, J., Murtagh, D. P., Stiller, G., and Walker, K. A.: Evolution and Variability of Water Vapour in the Tropical Tropopause and Lower Stratosphere Region Derived from Satellite Measurements, in: Advances in Atmospheric Science and Applications, vol. 708 of *ESA Special Publication*, p. 8, 2012.

Urban, J., Lossow, S., Stiller, G., and Read, W.: Another drop in water vapor, EOS Transactions, 95, 245 – 246, doi:10.1002/2014EO270001,
5    2014.

Vömel, H., Oltmans, S. J., Hofmann, D. J., Deshler, T., and Rosen, J. M.: The evolution of the dehydration in the Antarctic stratospheric vortex, Journal of Geophysical Research, 100, 13 919 – 13 926, doi:10.1029/95JD01000, 1995.

von Clarmann, T., Stiller, G., Grabowski, U., Eckert, E., and Orphal, J.: Technical Note: Trend estimation from irregularly sampled, correlated data, Atmospheric Chemistry & Physics, 10, 6737 – 6747, 2010.

10   Walker, K. A. and Stiller, G. P.: The SPARC water vapour assessment II: Data set overview, in preparation.





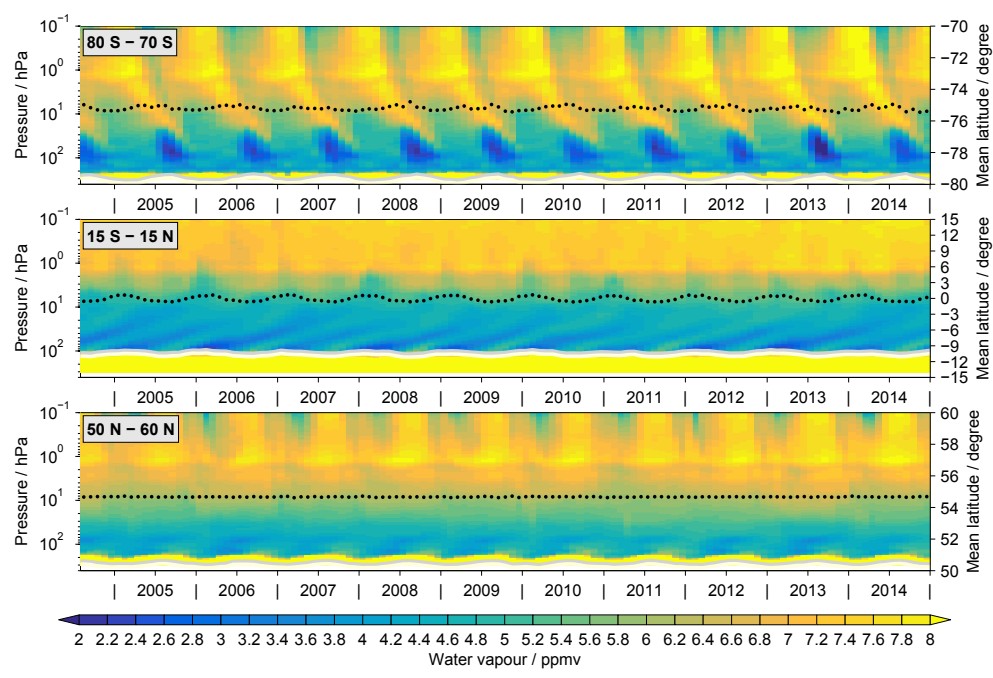

**Figure 1.** Water vapour time series for the latitude bands 80°S to 70° S (top panel), 15°S to 15°N (middle panel) and 50°N to 60°N (bottom panel) based on the MLS data. The light grey and white lines indicate the tropopause as derived from the MERRA reanalysis data. The black dots show the average latitude of the monthly mean data given on the right y-axis. White areas indicate that there are no data.





**Figure 2.** De-seasonalised time series at four different altitudes considering the latitude band $80°$ S to $70°$ S. In the legend the average latitude of the individual time series is indicated, which was calculated in two steps. First, for an individual monthly mean the latitudes of all profiles contributing to it were averaged. Any altitude dependence due to missing or screened data was ignored in this step. Finally, the mean latitudes over the entire time series were averaged. The same anomaly range (y-axis) has been used in all panels so that the differences in the anomaly and the spread is better comparable. On the x-axis the ticks are given in the middle of the year.



**Figure 3.** As Fig. 2, but considering the latitude band between 15°S and 15°N.



**Figure 4.** As Figs. 2 and 3, but here the time series for the latitude band between 50°N and 60°N are shown.





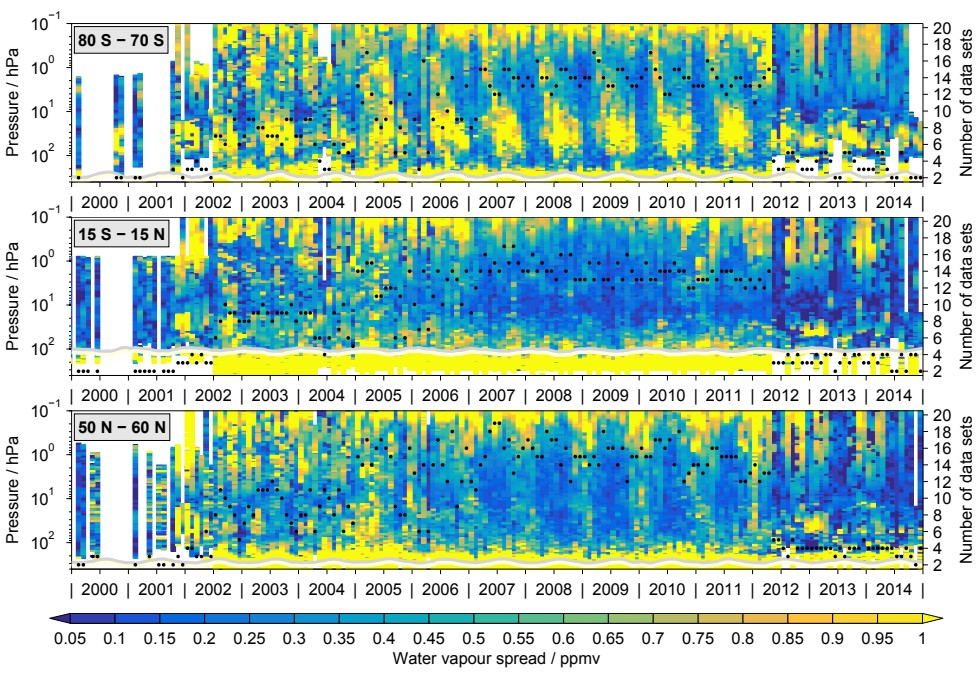

**Figure 5.** The difference between the maximum and minimum volume mixing ratio among the different de-seasonalised data sets as a function of time and altitude for the three latitude bands. The right axes and the corresponding black dots indicate the maximum number of data sets available for this analysis at a given time considering all altitudes.





**Figure 6.** Example correlations between de-seasonalised MIPAS-Oxford V5R NOM time series and those from other data sets. Results are only shown when the two data sets have an overlap of at least 12 valid monthly means. The dashed orange lines indicate the four altitudes for which the correlations between all data sets are shown in the following figures.





**Figure 7.** The correlations between de-seasonalised time series in the latitude band between 80° S to 70° S. The upper panel considers the 0.1 hPa (upper triangle) and 3 hPa (lower triangle) pressure levels, while in the lower panel the results at 10 hPa (upper triangle) and 80 hPa (lower triangle) are shown. Only data sets yielding any result at a given altitude are shown. Thus, the number of data sets can vary from altitude to altitude. Comparisons yielding no results are indicated by grey crosses. For comparisons with results (the coloured boxes) the number of months the two data sets actually overlap (i.e. both yield a valid monthly mean) are indicated.







**Figure 8.** As Fig. 7, but here the results for the latitude band between 15°S and 15°N are shown.





**Figure 9.** As Figs. 7 and 8, but considering the latitude band between 50°N and 60°N.



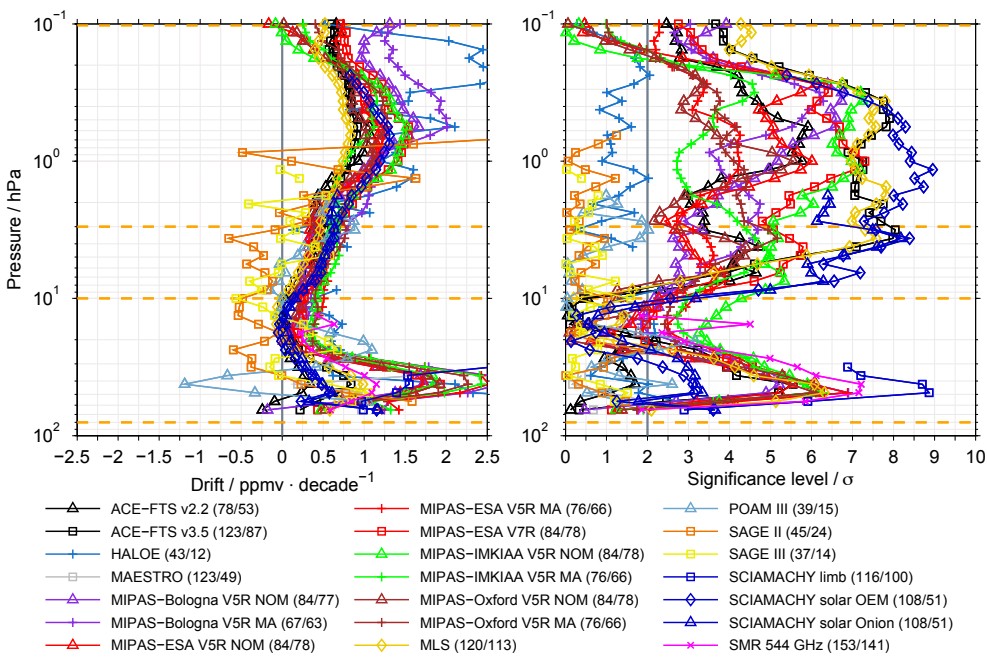

**Figure 10.** The left panel shows the drifts between the de-seasonalised time series of the SMR 489 GHz data set and the other data sets. In the right panel the corresponding significance levels of the drift estimates are shown and the $2\sigma$ level is marked by a vertical line. This example considers the latitude band between $50°$ N and $60°$ N. In the legend the first number indicates the overlap period (over all altitudes) of the two data sets, i.e. the time between the first and the last month both data sets yield a valid monthly mean. Results are only shown here when this time period is at least 36 months. The second number indicates during how many months both data sets actually yield a valid monthly mean.





**Figure 11.** Drifts between the different data sets in the latitude band between 80° S and 70° S at four specific altitudes. The drift estimates are based on the difference time series between the data sets given at the x-axis and the data sets given at the y-axis. Again, data sets are only shown if they yield any result at a given altitude. Besides the colour-coded drift estimates the result boxes contain additional information. In the upper left the overall time period the two data sets overlap is given first. The second number indicates how many months the data sets actually overlap. For a better visibility both significant and non significant drifts are marked. If a drift is not significant at the $2\sigma$ uncertainty level this is marked by a slant. If a drift is significant this is marked by a green frame and the significance level is noted in the lower right corner. In the colour bar, the drift is given in steps of 0.1 ppmv decade$^{-1}$ in the interval from $-1$ ppmv decade$^{-1}$ to 1 ppmv decade$^{-1}$. Outside this range the drift is given in steps of 0.5 ppmv decade$^{-1}$. 37





**Figure 12.** As Fig. 11, but here for the tropics, i.e. 15° S and 15° N.





**Figure 13.** As Figs. 11 and 12, but here the results for the latitude band between 50° N and 60° N are shown.



**Table 1.** Overview over the water vapour data sets from satellites used in this study.

| Instrument | Data set | Label | Number | Time period |
|---|---|---|---|---|
| ACE-FTS | v2.2 | ACE-FTS v2.2 | 1 | 03/2004 – 09/2010 |
|  | v3.5 | ACE-FTS v3.5 | 2 | 03/2004 – 12/2014 |
| GOMOS | LATMOS v6 | GOMOS | 3 | 09/2002 – 07/2011 |
| HALOE | v19 | HALOE | 4 | 10/1991 – 11/2005 |
| HIRDLS | v7 | HIRDLS | 5 | 01/2005 – 03/2008 |
| ILAS-II | v3/3.01 | ILAS-II | 6 | 04/2003 – 08/2003 |
| MAESTRO | Research | MAESTRO | 7 | 03/2004 – 12/2014 |
| MIPAS | Bologna V5H v2.3 NOM | MIPAS-Bologna V5H | 8 | 07/2002 – 03/2004 |
|  | Bologna V5R v2.3 NOM | MIPAS-Bologna V5R NOM | 9 | 01/2005 – 04/2012 |
|  | Bologna V5R v2.3 MA | MIPAS-Bologna V5R MA | 10 | 01/2005 – 04/2012 |
|  | ESA V5H v6 NOM | MIPAS-ESA V5H | 11 | 07/2002 – 03/2004 |
|  | ESA V5R v6 NOM | MIPAS-ESA V5R NOM | 12 | 01/2005 – 04/2012 |
|  | ESA V5R v6 MA | MIPAS-ESA V5R MA | 13 | 01/2005 – 04/2012 |
|  | ESA V7R v7 NOM | MIPAS-ESA V7R | 14 | 01/2005 – 04/2012 |
|  | IMKIAA V5H v20 NOM | MIPAS-IMKIAA V5H | 15 | 07/2002 – 03/2004 |
|  | IMKIAA V5R v220/221 NOM | MIPAS-IMKIAA V5R NOM | 16 | 01/2005 – 04/2012 |
|  | IMKIAA V5R v522 MA | MIPAS-IMKIAA V5R MA | 17 | 01/2005 – 04/2012 |
|  | Oxford V5H v1.30 NOM | MIPAS-Oxford V5H | 18 | 07/2002 – 03/2004 |
|  | Oxford V5R v1.30 NOM | MIPAS-Oxford V5R NOM | 19 | 01/2005 – 04/2012 |
|  | Oxford V5R v1.30 MA | MIPAS-Oxford V5R MA | 20 | 01/2005 – 04/2012 |
| MLS | v4.2 | MLS | 21 | 08/2004 – 12/2014 |
| POAM III | v4 | POAM III | 22 | 04/1998 – 11/2005 |
| SAGE II | v7.00 | SAGE II | 23 | 01/1986 – 08/2005 |
| SAGE III | Solar occultation v4 | SAGE III | 24 | 04/2002 – 06/2005 |
| SCIAMACHY | Limb v3.01 | SCIAMACHY limb | 25 | 08/2002 – 04/2012 |
|  | Lunar occultation v1.0 | SCIAMACHY lunar | 26 | 04/2003 – 04/2012 |
|  | Solar occultation - OEM v1.0 | SCIAMACHY solar OEM | 27 | 08/2002 – 08/2011 |
|  | Solar occultation - Onion peeling v4.2.1 | SCIAMACHY solar Onion | 28 | 08/2002 – 08/2011 |
| SMILES | NICT v2.9.2 band A | SMILES-NICT band A | 29 | 01/2010 – 04/2010 |
|  | NICT v2.9.2 band B | SMILES-NICT band B | 30 | 01/2010 – 04/2010 |
| SMR | v2.0 544 GHz | SMR 544 GHz | 31 | 11/2001 – 12/2014 |
|  | v2.1 489 GHz | SMR 489 GHz | 32 | 11/2001 – 08/2014 |
| SOFIE | v1.3 | SOFIE | 33 | 08/2007 – 09/2014 |