# Peer review of "The SPARC water vapour assessment II: Comparison of stratospheric and lower mesospheric water vapour time series observed from satellites"

_Atmospheric Measurement Techniques, 2018_

## Referee Comment (RC1) · Anonymous Referee #1 · 4 Apr 2018

General Comments

This manuscript presents useful analyses regarding intercomparisons of H2O time series from many satellite measurements, in terms of monthly means, data spread, correlation coefficients, and drifts between the time series, as part of the SPARC WAVAS-II assessment. The methods are generally sound, although some aspects could/should be clarified, and some plots take a while to digest, as there are many curves and data sets to consider, which also makes a clear/useful summary somewhat difficult to present. Different readers or investigators may proceed differently in terms of how to

use or discard certain data sets, based on these sorts of results. I find that (at least) more caveats regarding the error bars and significance levels would be useful, given that autocorrelation effects are ignored. I also find that the conclusion regarding H2O data usage is fairly bland and "politicially correct", but not very useful scientifically, since some data sets clearly exhibit more outliers or drifts than others. Such issues are not easy to deal with in terms of trying to assess the trends in H2O, ultimately, and this manuscript does not try to guide the reader in that direction, which is maybe alright for a more limited and well-defined scope in this paper. This seems to imply that everyone should try to draw their own conclusions about how to decide, which may indeed be better than trying to impose only one particular solution, and there would be a lot of extra work involved to make these sorts of assessments. This work may well be followed, in time, with more details in a related manuscript (mentioned in this work) or by other work on H2O trend assessment, and this manuscript should stand on its own as well. I include some suggestions for improvements in the specific comments below; there are also a few statements where not enough information is provided. Overall, this work will provide benefits to investigators of global stratospheric H2O; after some changes based on the suggestions below (ranging from minor to somewhat more major), it could be made (even) more suitable for publication, and I would want to see it published (my recommendation really stands somewhere between "only minor" and "some more major" revisions).

Specific Comments (some in between minor and major)

1. Impact of different vertical (and horizontal) resolutions: I realize that this is a somewhat difficult topic to deal with, but one should at least acknowledge that these differences between measurement systems can lead to differences in the time series (which are then interpolated to a fine grid); for example, a tape recorder signal will not look exactly the same to different instruments. If you could touch on this for some of the highest and lowest resolution instruments (ignoring the horizontal component), and comment on whether you think there are impacts for these results, this would enhance

the paper quality. Also, indicating the actual "canonical" vertical resolution for a typical stratospheric measurement (e.g., under the instrument names in Table 1, or by adding a column to this Table), would provide useful information that the reader does not have to try to fetch elsewhere, or remember. There are enough assumptions made already about the readers' knowledge of each instrument system (even regarding first-order information like vertical resolution).

2. Impact of known issues in certain measurements: There is not quite enough discussion, in my view, of certain known issues that could play a significant role in these intercomparisons. Each instrument team representative could (have) provide(d) more feedback on actual knowledge of instrument degradation issues or known drifts. For example, there have been issues with drifts in MIPAS ozone in the literature (which are touched on briefly in the context of some of the MIPAS ESA versions), and what about H2O? A clearer summary, up front in the drift section for example, regarding what retrieval versions have some correction and which ones do not, would be useful. There has also been some evidence for drifts in the MLS measurements versus sonde data (as presented by Hurst et al.); is this detectable in the plots or drifts you come up with here? Also, is there another part of this WAVAS assessment that attempts to consider drifts with respect to ground-based measurements, and would that not be a good cross-reference to consider, if so? (If there is not, that will be work for the future, I reckon - and these assessments undoubtedly take up a lot of time and work, I am aware of this, not trying to downplay the useful work that has been done already).

3. Some of the drifts are quite large, and each instrument's results tend to get a bit buried in the sea of curves (e.g. in Figs 2 through 4, and Figure 10; showing Figure 10 for all latitude bins could also be helpful, in the main text, even if Figs. 11 through 13 are quite nice, but somewhat less easy to grasp than Fig. 10. One also wishes for a more quantitative summary of the accuracy of expected trends based on the "combination" of knowledge shown here, even without getting into the trends themselves. The spread between the curves in Figure 10 could be such a measure, say in percent/decade

rather than ppmv/decade (just a personal preference but not critical since the vertical gradients in H2O are not as strong as for O3, for example). First, one might want to eliminate some of the outliers (e.g. despike the drift results using techniques you have used with the MAD for example, or some 4 or 5 sigma type of screening), and then calculate the rms spread versus pressure. One could also superpose three curves (one for each latitude bin you have considered). In fact, in an ideal world, showing this for even more latitude bins to check for consistencies or inconsistencies (and systematic effects for certain instruments) would provide an even more complete picture (such as in a latitude/pressure contour plot). As comprehensive as this work looks already, there is even more information to go after, as you imply in reference to other manuscript(s) in preparation or in press - and this is not something that is a serious flaw, as long as there is some level of consistency between the latitude bins chosen here (if not, then the reader can conclude that more work is really needed to try to make sense of all these measurements). Even a 0.5 ppmv/decade drift is fairly large, since this is about 10% and the trends in H2O (or expectations for long-term trends) are not larger than this.

4. Your final statements (in the Abstract and in the Conclusions) about being able to consider "all data sets... when data set specific characteristics (e.g. a drift) and restrictions (e.g. temporal and spatial coverage) are taken into account" seem to be too much of a "politically correct" stretch, even though I realize that this is often done to please every team making up an assessment-type paper. What is really required (besides a lot of work) to try to actually take such effects into account and really assess trends in H2O (as is done for ozone)? This is certainly missing from this work - precisely because this is a lot easier said than done. I will not try too hard to force a different consensus view or statement, but it is something to reconsider, I would argue, in terms of what the best message for readers really is, as a scientific statement about the uncertainties and possibilities, given the large spreads or at least, the existence of several outliers. How is one supposed to "consider" known drifts, or the large spread in (some of) these results? Either they all get characterized better versus ground-based

data (if and where possible), or versus some "cleaner" average satellite time series - or some other solution, if a more satisfying recommendation can be pursued. At the very least, I am asking for some thought about this and an attempt to make a more useful statement, even if this may just be a suggestion in order to arrive at a hopefully robust consensus about the state of the trends in H2O. I am not asking that all that work be done for this manuscript, just for a better suggestion than "use everything but consider everything", which is basically not providing any useful recommendation in terms of specifics. If it really is too difficult to arrive at a better (consensus) statement, saying that this was attempted is still better than just leaving the vague conclusion in as it stands now; hopefully this stimulates a bit more discussion, at the very least (again, without requiring a lot of detailed work). This may have been a point of discussion already among co-authors.

5. It is stated that the error bars (and drift) estimates do not take into consideration in the regressions (see the statement near the top of page 6). I assume that this is also the case for the drift estimation (calculated via a simple linear trend model applied to the time series of differences). Since a lot of the results depend on the significance level, the underestimated set of error bars, which is typically the result of the neglect of autocorrelation effects, will imply that somewhat erroneous conclusions are arrived at whenever you discuss significance levels. This could often be a non-negligible effect indeed. The trend estimates are not likely to change very much, and this is more of an impact on the error bars themselves, which means that you would have more slant lines across more boxes in Figures 11-13, and Figure 10 would also be affected (mainly for the panel on the right). I realize that this is also a lot of work to try to do rigorously, so any rough estimate of the impact of this (for an example, not for all the series) would be a useful addition to the work already done, mainly as a comment, not necessarily in terms of changing the Figures themselves. One could comment, for example, that it is likely that most (or almost all) of the boxes would end up being non-statistically-significant (this is apparently already the case actually). This does not invalidate the fact that there are outliers in an often systematic way, and that the drifts

do show relative inter-instrument effects and certain tendencies, even if not statistically significant. So I still think that the flavor of (and interest in) the results can be preserved, despite the fact that there is a lack of full rigor in the treatment of statistical significance and some of the conclusions. Some sort of statement that goes slightly beyond merely stating that this effect is completely neglected would be useful for readers, since this is a mathematically known issue (that often gets ignored). If you can prove that this is an insignificant omission, please show this with further analyses (although I personally do not believe that this is the case, without more investigation). If this is completely ignored, however, even if one states that it is being ignored, the reader will get the (incorrect) impression that the results of significance are to be taken at face value, which is really not the case. A more cautionary note is what I am mainly after here, since these issues are too often ignored altogether, but not a complete rework using different statistical methods (e.g., a bootstrap method could also be applied for error estimates).

6. You sometimes state that noisy measurements are a cause for poorer results, for example in the correlations. You have also used the different impact of clouds as an explanation of some differences. I am not convinced that these explanations are well enough proven, at least by what I see in this manuscript. When you have monthly means, most of the results will have very small standard errors in the means, and noise itself becomes much less of an issue. This can be demonstrated for each instrument, and some will be more "noisy" than others, this is still true, depending on the number of data points (and the actual single-profile noise). I would urge you to more carefully consider those comments and convince yourselves of certain cases where these statements are made. If you are not convinced, or convincing, maybe you should invoke unknown systematic effects as well, as another option that could also play a role (when one wants to try to provide a range of options for some differences without a more complete investigation, which could take a while for the multitude of data sets you are considering here, I agree). Alternatively, if you do have a few good examples, you could add supplementary material to show differences of time series

with the noise values (for example), or something else relating to cloud studies (or another reference, possibly). I also tend to think that sampling will play a bigger role than one might think in some of the differences (for example in the issues mentioned in the paragraph before Section 4.4), even if you already do mention some examples of this issue. I just want to avoid statements that sound like careful work has gone into the conclusion, with little proof given; while there can be (is) some level of trust regarding work not shown in the paper, I am not entirely convinced that all the possible explanations have been vetted enough. One can try to investigate and comment about a few of the more obvious cases with poorest results, for example, at certain pressures or latitudes. It is also somewhat surprising to see some of the differences between the various MIPAS retrievals, in terms of how some of the larger differences can come about. Again, the main flavor of the results will most likely not change, and it would be a large undertaking to try to resolve "everything", so I (and others) will need to read this as the way things are, for now, not that one shouldn't expect better agreement in the end, given enough time, etc... but there are still a number of unresolved issues. Some of the writing gives a lot of description of differences, with maybe not enough "potential explanations", and we can infer that some things are not understood yet, which is not completely unexpected either. This does not make this work unworthy of publication, in my view, although one would prefer to see a lot of these uncertainties reduced, or somewhat better explained.

Minor Comments

- pg. 2, L3, change ratio to ratios.

- pg. 2, L8, change :when data" to "if data", although my specific comments are not that positive regarding this sort of statement.

- pg. 3, L18; I would suggest something shorter/better like "with water vapour abundances recovering after 2004-2005".

- pg. 4, Eq. (1), interesting use of "z" for pressure, rather than "p", but this is just a

personal preference, nothing to really change.

- pg. 5, lines 8 and 11, there is a change in tense (from "was calculated" to "are discarded"); I would recommend using the same tense in general, inasmuch as possible (e.g., "were discarded").

- pg. 5, L15, maybe change "refined" to "relaxed", or "less stringent criterion".

- pg. 6, L11, typo in "alltogether" [altogether].

- pg. 6, L12, change "ratio" to "ratios".

- pg. 6, L22, delete "the" before "the".

- pg. 6, L26, change "was 3.6" to "removed 3.6".

- pg. 6, L29, add a comma after "measurements".

- pg. 7, L1, change "result" to "results".

- pg. 9, L18, are compared qualitatively.

- pg. 10, L26, delete "and" before "2011".

- pg. 11, L14, available for these altitude and latitude regions.

- pg. 11, L18, change "as those" to "than those".

- pg. 11, L33, change "in order" to "on order".

- pg. 12, L14-15, in the other data sets, anomalies up to only 0.4-0.8...

- pg. 12, L17, anti-correlated with the time series

- pg. 12, L23, add a comma after V5H.

- pg. 13, L1, are found during 2004-2008, when SAGE II...

- pg. 13, L9, change "regions" to "region".

- pg. 13, L17, change slightly decreasing to decreasing slightly.

- pg. 13, L21, 22, there is a lot of repetition of the same references to the drops in H2O.

- pg. 14, L17, change "coefficient" to "coefficients", and one line lower, change "is" to 'are".

- pg. 14, L20, change "in form" to "in the form" or to "as".

- pg. 14, L30, change "NOM than" to "NOM as".

- pg. 14, L31, most data sets between 1 and 30 hPa.

- pg. 15, L9, change "varies" to "vary".

- pg. 15, L19, "the number of months of overlap between time series is given..."

- pg. 15, L25, the number of overlap months is not that high...

- pg. 16, L2, the number of overlap months is rather low (same for L5/6, and L10, and L20 on pg. 17).

- pg. 16, L7, overview of the temporal...

- pg. 16, L11, An example of a negative correlation, despite a high number of overlapping months, is for the correlation ...

- Also on this page, it is not very surprising when ACE-FTS data versions or MIPAS versions correlate well... it is the opposite hat is more surprising.

- pg. 16, L23, which "both quantities" are you talking about here, please clarify.

- pg. 17, L14, change "were both data sets" to "for which both data sets".

- pg. 17, L19, change "at the" to "on the" (x-axis and y-axis).

- pg. 18, L17, drifts are significant in most cases.

- pg. 18, L26, change "pattern" to "patterns" [are...].

- pg. 18, L34, Exceptions are HIRDLS... and MAESTRO...

- pg. 19, L25, change "because" to "that".

- pg. 19, L27, influenced differently by clouds.

- pg. 19, L30/31, Larger deviations in the lower mesosphere occur in the case of the MIPAS NOM data sets, which are close to their upper retrieval limit there, and thus more uncertain.

- pg. 20, Since the dehydration is only partly a seasonal oscillation,...

- pg. 20, L8, were also assessed; this indicates if the longer-term variations (trends) ...

- pg. 20, L11, change "level" to "levels".

- pg. 20, L11, The most significant drifts were found in the tropics, where there is low..., which...

- pg. 20, L13, Drifts were also calculated...

- pg. 20, L22, delete "amongst others".

- pg. 20, L32, that in addition to correcting...

- pg. 20, L33, changes in the calibration were madew within the HIRDLS mission...

- pg. 20, L34, data sets encounter large uncertainty...

- References, there are a few refs. with no doi numbers (Randel, Remsberg, von Clarmann).

- pg. 28, bottom line, and the spread are more easily compared.

- Figure 5, one could also find an rms diagnostic versus pressure and contract the time axis this way, as an additional Figure that could overlay the three latitude regions, so as to get a maybe better overview of this quantity over pressure and latitude (and some of the colors are not so easy to differentiate).

- Figure 10, last sentence, Th second number indicates the number of months for which both data sets ...

- Figure 11, line 2, change "at" to "on" (x-axis, y-axis). Line 4, upper left the overall overlap period between data sets is given. The second number indicates for how many months ...

---

## Referee Comment (RC2) · H. C. Pumphrey (Referee) · 3 May 2018

**General comments**

This is a useful summary paper, which, for the most part, presents a large and complex body of information in a digestible manner. Most of the items that make it difficult for the reader are related to the large number of datasets from a single instrument (MIPAS). If it were possible to do anything to separate the inter-MIPAS comparisons from the

(more important) comparisons between different instruments, then I would like to see this done. But I recognise that this might be too large a change to be made.

Like referee 1, I am conscious that this paper does not provide any kind of guidance to the reader as to which data sets are the most useful. I imagine that this is deliberate and is done to avoid annoying any of the data providers. I nevertheless feel that some sort of opinion as to which datasets are the most useful for which purposes would not be out of place.

**Specific comments**

- Page 4 line 10: The authors note that the data from UARS MLS are not considered. I do not think these data would add much as there are less than 18 months worth. But the authors have included the ILAS-II and SMILES data, which cover even shorter time periods, so I think they should explain why they are including ILAS-II and SMILES, but not including UARS MLS. (**Disclaimer:** I am responsible for the UARS MLS water vapour data.)

- Figure 1: The labelling of the colour bar is rather cluttered; it might be preferable to label only 2,3,4,5,6,7, and 8 ppmv.

- Page 8 line 14: I would remove the words "(contour time series)" as the data are presented as an image — no contours have been drawn.

- Page 8 line 27: Again (and in several subsequent places, including in the supplement), remove the word "contour" as figures 1, 5, and S1-S3 contain no contours.

- Figure 2-4: I do rather wish that the various teams involved with MIPAS would agree on one best product. Half of the products shown in these figures are from this one instrument. I understand that the instrument has various operating

modes which are not directly comparable, so a single product may not be practical. But 13 different products are very confusing for the reader and the data user. It might have been preferable to first form some sort of combined or approved MIPAS dataset (or, at most, one for each operating mode) to be compared to other instruments. I do not imagine that the authors will want to re-design the entire paper along these lines. But for the purposes of these figures it might be better to show only one MIPAS dataset (and possibly, only one ACE-FTS data set — why do we need V2.2 if V3.5 is supposed to be an improvement?).

- Page 10-12: Many of the features of the data described here are rather hard to see in figures 2-4, on account of the large number of lines. I am not sure what to suggest (other than not showing all the MIPAS data!).

- Figure 5: The black dots are very difficult to see, especially against the darker end of the colour scale. Potential solutions include joining the dots with a line and/or using a colour (red?) which does not form part of the colour scale.

- Page 14 lines 1-15: One of the most striking features of the figure is the change in 2012 caused by the end of the Envisat mission and hence of the myriad MIPAS datasets. It strikes me that the use of the max-min difference to quantify spread means that this plot mostly tells you about where the noisiest dataset is at its noisiest. I have to question whether this is the most useful measure of either atmospheric variability or overall data quality.

- Figures 7-9: These figures are an interesting way of showing a large amount of summary information in a clear way, and in a small space. Something that caused me a bit of confusion was the way that the numbers in the upper triangle do not always align with those in the lower triangle. This is because different levels have different datasets available. It might be worth inserting blank rows into one or other triangle in each pane so that the two triangles have the same numbering scheme.

- Figures 11-13: In addition to the suggestion I make regarding figures 7-9, figures 11-13 have text on them which is VERY small. It is commonly recommended that text on a figure should be no smaller than the figure caption text in the final typeset version of the article. There is clearly a bit of leeway on this recommendation, but the text on this figure is so tiny that it is very annoying for the reader, especially for middle-aged readers who are still cross that they need reading glasses. I am not sure what to suggest here, because simply making the text bigger will not work: in some cases it is already impinging on the diagonal lines.

- Page 21: dedication. I too have good memories of working briefly with Jo Urban, and was saddened to hear of his passing at such a young age.

**Technical corrections**

- Page 2 line 1: "allowed considering the time period" reads rather oddly. Maybe write "allowed us to consider the time period" or "allowed the consideration of the time period".

- Page 3 line 17: "One drop (also known as the millennium drop) . . . " The "also" does not read right as you have not first given another name by which the drop is known. Maybe write "One drop (sometimes known as the millennium drop) . . . ".

- Page 11 Line 25: remove comma after "Both"

- Page 14 line 30: replace "than" with "as"

---

## Author Comment (AC1) · 25 Jun 2018

We thank referee 1 for the constructive, helpful criticism and the suggestion for revision. We followed the suggestions of referee 1 and revised the manuscript accordingly.

*General Comments*
*This manuscript presents useful analyses regarding intercomparisons of H2O time series from many satellite measurements, in terms of monthly means, data spread, correlation coefficients, and drifts between the time series, as part of the SPARC*

[Figure]

*WAVAS-II assessment. The methods are generally sound, although some aspects could/should be clarified, and some plots take a while to digest, as there are many curves and data sets to consider, which also makes a clear/useful summary somewhat difficult to present. Different readers or investigators may proceed differently in terms of how to use or discard certain data sets, based on these sorts of results. I find that (at least) more caveats regarding the error bars and significance levels would be useful, given that autocorrelation effects are ignored.*

**This is a misunderstanding. Autocorrelation has been considered. See our answer below.**

*I also find that the conclusion regarding $H_2O$ data usage is fairly bland and "politically correct", but not very useful scientifically, since some data sets clearly exhibit more outliers or drifts than others. Such issues are not easy to deal with in terms of trying to assess the trends in $H_2O$, ultimately, and this manuscript does not try to guide the reader in that direction, which is maybe alright for a more limited and well-defined scope in this paper.*

**This study is not about assessing trends or giving guidance on how to derive trends, but on assessing the differences of the time series derived from satellites. The here derived results will of course be of help for future studies focusing on trends studies since data set specific characteristics and problems that need to be considered or could make a trend estimation difficult are already revealed.**

*This seems to imply that everyone should try to draw their own conclusions about how to decide, which may indeed be better than trying to impose only one particular solution, and there would be a lot of extra work involved to make these sorts of assessments. This work may well be followed, in time, with more details in a related manuscript (mentioned in this work) or by other work on $H_2O$ trend assessment, and this manuscript should stand on its own as well.*

**This manuscript is part of the WAVAS II special issue and joins a number of papers dealing with other aspects of the quality assessment. Our intention**

**indeed was not to impose a particular solution on which data set to use, but to provide a thorough assessment of the water vapour time series derived from satellites. Based on the specific application the users need to decide which data set is best suited for this purpose. We provide now some more guidance on this as given in our answer below (answer to referee comment 4).**

*I include some suggestions for improvements in the specific comments below; there are also a few statements where not enough information is provided. Overall, this work will provide benefits to investigators of global stratospheric $H_2O$; after some changes based on the suggestions below (ranging from minor to somewhat more major), it could be made (even) more suitable for publication, and I would want to see it published (my recommendation really stands somewhere between "only minor" and "some more major" revisions).*

**We have revised the manuscript according to the suggestions given in the general and specific comments. Our detailed answers to the points of criticism raised in the general comments are as follows: For the de-seasonalisation of the time series we have indeed not considered any auto-correlation in the regression analyses. This has been a compromise in favour of the more sparse data sets where the regression occasionally did not converge. For the drift analyses, however, autocorrelation has of course been considered to get the optimal uncertainty estimates. In the manuscript this is clearly stated in Sections 3.1 and 3.3, respectively. On P6, L3 in the AMTD manuscript (P8, L5 in the revised manuscript) it is stated:** *Autocorrelation effects and empirical errors (Stiller et al., 2012) were not considered in this regression.* **On P6, L11 in the AMTD manuscript (P8, L14 in the revised manuscript) we state:** *Here, unlike in the regression for the de-seasonalisation, auto-correlation effects and empirical errors were considered to derive optimal uncertainty estimates for the drift.* **Regarding the comment on our conclusion: We do not want to impose one particular solution, but of course we want to give some kind of guidance. Therefore, we improved our concluding remarks as given below in the answer to**

**the specific comment on this issue (answer to referee comment 4). Further, we once again would like to point out that this study is meant as a stand-alone study as part of a larger activity of which the results are all summarized in a special issue. Thus, not every information from the other papers can be repeated here. Therefore, we ask the readers to collect the additional information from the respective publications in the special issue.**

*Specific Comments (some in between minor and major)*
*1. Impact of different vertical (and horizontal) resolutions: I realize that this is a somewhat difficult topic to deal with, but one should at least acknowledge that these differences between measurement systems can lead to differences in the time series (which are then interpolated to a fine grid); for example, a tape recorder signal will not look exactly the same to different instruments. If you could touch on this for some of the highest and lowest resolution instruments (ignoring the horizontal component), and comment on whether you think there are impacts for these results, this would enhance the paper quality. Also, indicating the actual "canonical" vertical resolution for a typical stratospheric measurement (e.g., under the instrument names in Table 1, or by adding a column to this Table), would provide useful information that the reader does not have to try to fetch elsewhere, or remember. There are enough assumptions made already about the readers' knowledge of each instrument system (even regarding first-order information like vertical resolution).*

**We do not assess here the "contour" time series, but the de-seasonalised time series. Hence, the tape recorder is not relevant here. For the "contour" time series the vertical resolution will of course influence the tape recorder signal. A discussion on this influence can be found in Lossow et al. (2017). An overview of the vertical resolutions of the WAVAS data sets will be provided in the WAVAS data set characterisation paper by Walker et al., in preparation. This will be the central place for this kind of information and, thus, not be repeated here. We refer to the study by Walker et al. at several places in the manuscript. For the**

[Figure]

**horizontal resolution we have no such summary and most data sets even do not do an assessment of the actually retrieved horizontal resolution, but only of the horizontal sampling. It should be kept in mind that these two entities are not the same. Sampling biases of course play a role for our analyses but are unavoidable. We added the following text to the "Summary and Conclusion":** *There are multiple reasons that give rise to the observed differences between the individual data sets. A thorough discussion on this is given in Lossow et al. (2017). From this study we know that the most important contributions arise from differences in temporal and spatial sampling, the influence of clouds or NLTE effects. Other reasons include systematic differences, for example calibration problems. However, for the time series comparison we would rank sampling biases as well as systematic errors as the most important reason for the differences as was discussed by Toohey et al. (2013) based on trace gas climatologies.*

*2. Impact of known issues in certain measurements: There is not quite enough discussion, in my view, of certain known issues that could play a significant role in these intercomparisons. Each instrument team representative could (have) provide(d) more feedback on actual knowledge of instrument degradation issues or known drifts. For example, there have been issues with drifts in MIPAS ozone in the literature (which are touched on briefly in the context of some of the MIPAS ESA versions), and what about $H_2O$? A clearer summary, up front in the drift section for example, regarding what retrieval versions have some correction and which ones do not, would be useful. There has also been some evidence for drifts in the MLS measurements versus sonde data (as presented by Hurst et al.); is this detectable in the plots or drifts you come up with here? Also, is there another part of this WAVAS assessment that attempts to consider drifts with respect to ground-based measurements, and would that not be a good cross-reference to consider, if so? (If there is not, that will be work for the future, I reckon - and these assessments undoubtedly take up a lot of time and work, I am*

*aware of this, not trying to downplay the useful work that has been done already).*

**The purpose of this study is to assess the quality of water vapour time series derived from satellite observations. Thereby, our intention is to detect issues like drifts between data sets from a systematic, consistent and independent assessment. If we find specific issues here that have not documented before as e.g. the drift of Odin/SMR, then such issues are clearly stated since this is the purpose of our study. On the other hand, if we find issues and these are consistent with earlier findings published in the literature we refer to the respective studies. For the drift in MIPAS or the drift in SMR, there are no published studies on these drifts. Therefore, our study adds new, previously not available information. The Walker et al., in preparation, paper will be the main data set reference with all knowledge before this assessment being included, like e.g. the MLS drift vs. FPH or the drift in MIPAS V5. It is not useful to add in all WAVAS papers this information over and over again unless we see it as quite important for the current study. This is e.g. the case for Table 1 which provides an overview over the water vapour data sets from satellites used in this study and can also be found in Lossow et al. (2017). We know that it is a clear drawback for all studies currently under revision that the Walker et al. paper is not published or even submitted yet. So therefore we have no other choice to than to ask the referee and readers for more patience for these kind of additional information (as stated above, in case the information is necessary for the specific study, it will be repeated).**

*3. Some of the drifts are quite large, and each instrument's results tend to get a bit buried in the sea of curves (e.g. in Figs 2 through 4, and Figure 10; showing Figure 10 for all latitude bins could also be helpful, in the main text, even if Figs. 11 through 13 are quite nice, but somewhat less easy to grasp than Fig. 10. One also wishes for a more quantitative summary of the accuracy of expected trends based on the "combination" of knowledge shown here, even without getting into the trends*

*themselves. The spread between the curves in Figure 10 could be such a measure, say in percent/decade rather than ppmv/decade (just a personal preference but not critical since the vertical gradients in $H_2O$ are not as strong as for $O_3$, for example). First, one might want to eliminate some of the outliers (e.g. despite the drift results using techniques you have used with the MAD for example, or some 4 or 5 sigma type of screening), and then calculate the rms spread versus pressure. One could also superpose three curves (one for each latitude bin you have considered). In fact, in an ideal world, showing this for even more latitude bins to check for consistencies or inconsistencies (and systematic effects for certain instruments) would provide an even more complete picture (such as in a latitude/pressure contour plot). As comprehensive as this work looks already, there is even more information to go after, as you imply in reference to other manuscript(s) in preparation or in press - and this is not something that is a serious flaw, as long as there is some level of consistency between the latitude bins chosen here (if not, then the reader can conclude that more work is really needed to try to make sense of all these measurements). Even a 0.5 ppmv/decade drift is fairly large, since this is about 10% and the trends in $H_2O$ (or expectations for long-term trends) are not larger than this.*

**We consider these suggestions to be beyond the scope of the current (already quite comprehensive) study. Our focus is on comparing the time series and not to provide an estimation of trends or errors in the trend estimates. These can be done in follow up studies. Further, as discussed for example in Lossow et al. (2018a) it is inevitable to first understand the differences between the time series for being able to yield consistent trend estimates. Regarding the spread in Figure 10: We do not think this is a really helpful measure. First of all the different lines are based on estimates for different time periods. Hence, the spread has at least partly natural reasons that we cannot separate out. Further, the drifts presented in Figure 10 are only relative to one data set, namely SMR, which we picked as example data set because it has an obvious drift. A more general assessment combining all results will be provided in Lossow et al. (2018b), in**

preparation. We think it is not a good idea to give the drift in percent/decade instead of ppmv/decade. First of all, we look here at the trend component of the difference time series derived from de-seasonalised data. Second, there are clear biases in the absolute data which clearly influence the relative estimate. Therefore, using absolute estimates rather than relative estimates seems to be the best option for our study. There is also no point in doing this analyses for more latitude bands. We have picked three latitude bands which cover the major climatic regions and give the best overview over the results. Considering more latitude bands would make this already quite comprehensive study even more comprehensive. Further, one does not gain much more insight from showing Figure 10 for all three latitude bands considered in this study (see Figure 1 in this reply) since quite similar results are derived. The paper is much more concise for just showing one latitude band as example (as it is done presently in Figure 10). It is correct that we derive drifts beyond 0.5 ppmv, but in these cases the overlap of the time series is mostly not that long (minimum overlap period is just 36 months) or these drifts are not significant. Further, to put our results in relation to other results: the trend differences between the FPH observations in Boulder and the merged satellite time series for the time period from the late 1980s until 2010 are also as large as 0.5 ppmv/decade.

*4. Your final statements (in the Abstract and in the Conclusions) about being able to consider "all data sets... when data set specific characteristics (e.g. a drift) and restrictions (e.g. temporal and spatial coverage) are taken into account" seem to be too much of a "politically correct" stretch, even though I realize that this is often done to please every team making up an assessment-type paper. What is really required (besides a lot of work) to try to actually take such effects into account and really assess trends in $H_2O$ (as is done for ozone)? This is certainly missing from this work - precisely because this is a lot easier said than done. I will not try too hard to force a different consensus view or statement, but it is something to reconsider, I would argue,*

*in terms of what the best message for readers really is, as a scientific statement about the uncertainties and possibilities, given the large spreads or at least, the existence of several outliers. How is one supposed to "consider" known drifts, or the large spread in (some of) these results? Either they all get characterized better versus ground-based data (if and where possible), or versus some "cleaner" average satellite time series - or some other solution, if a more satisfying recommendation can be pursued. At the very least, I am asking for some thought about this and an attempt to make a more useful statement, even if this may just be a suggestion in order to arrive at a hopefully robust consensus about the state of the trends in $H_2O$. I am not asking that all that work be done for this manuscript, just for a better suggestion than "use everything but consider everything", which is basically not providing any useful recommendation in terms of specifics. If it really is too difficult to arrive at a better (consensus) statement, saying that this was attempted is still better than just leaving the vague conclusion in as it stands now; hopefully this stimulates a bit more discussion, at the very least (again, without requiring a lot of detailed work). This may have been a point of discussion already among co-authors.*

**Instead of stating that "all" data sets can be used for studying atmospheric water vapour variability and trends we state now that this is the case for "most" data sets. We think this is a realistic statement. In fact, it is quite difficult to judge which data set is the best one to use. That simply depends on the scientific application and on which altitude region or latitude region the study is focused on. For trend analyses the longest data sets with the highest spatial and temporal coverage have of course, for such kind of studies, a clear advantage. We changed the last paragraph of the conclusion as follows:** *Nevertheless, although the water vapour data sets have been thoroughly assessed in this study it is difficult or rather impossible to judge on which data set is the best one to use for future modelling and observational studies. This simply can only be answered with respect to the specific science application the data set should be used for. For future studies on e.g. water vapour trends we can state*

*that the data sets that provide the longest measurement record with a high spatial and temporal coverage have an advantage over the ones which provide only observations in specific latitude bands and/or altitude regions. For data sets that have a drift relative to other data sets as e.g. SMR 489 GHz, the drift has to be taken into account and data sets that are simply too short (less than one year) as e.g. ILAS-II and SMILES cannot be used for trend studies at all. Once again, we need to point out here that this study is not about trends, so therefore we will not provide any trend assessments in this study. Likewise, we have to point out that an average time series (as the multi-instruments mean by Hegglin et al., 2013) has never been considered as a subject for this study as well since there are also caveats concerning the usage of an average time series.*

*5. It is stated that the error bars (and drift) estimates do not take into consideration in the regressions (see the statement near the top of page 6). I assume that this is also the case for the drift estimation (calculated via a simple linear trend model applied to the time series of differences). Since a lot of the results depend on the significance level, the underestimated set of error bars, which is typically the result of the neglect of autocorrelation effects, will imply that somewhat erroneous conclusions are arrived at whenever you discuss significance levels. This could often be a non-negligible effect indeed. The trend estimates are not likely to change very much, and this is more of an impact on the error bars themselves, which means that you would have more slant lines across more boxes in Figures 11-13, and Figure 10 would also be affected (mainly for the panel on the right). I realize that this is also a lot of work to try to do rigorously, so any rough estimate of the impact of this (for an example, not for all the series) would be a useful addition to the work already done, mainly as a comment, not necessarily in terms of changing the Figures themselves. One could comment, for example, that it is likely that most (or almost all) of the boxes would end up being non-statistically-significant (this is apparently already the case actually). This does not invalidate the fact that there are outliers in an often*

*systematic way, and that the drifts do show relative inter-instrument effects and certain tendencies, even if not statistically significant. So I still think that the flavor of (and interest in) the results can be preserved, despite the fact that there is a lack of full rigor in the treatment of statistical significance and some of the conclusions. Some sort of statement that goes slightly beyond merely stating that this effect is completely neglected would be useful for readers, since this is a mathematically known issue (that often gets ignored). If you can prove that this is an insignificant omission, please show this with further analyses (although I personally do not believe that this is the case, without more investigation). If this is completely ignored, however, even if one states that it is being ignored, the reader will get the (incorrect) impression that the results of significance are to be taken at face value, which is really not the case. A more cautionary note is what I am mainly after here, since these issues are too often ignored altogether, but not a complete rework using different statistical methods (e.g., a bootstrap method could also be applied for error estimates).*

**No, this is a misunderstanding. See our answer to the general comment. Autocorrelation and empirical errors are considered. The regression follows the method by Stiller et al. (2012) as clearly stated in the manuscript.**

*6. You sometimes state that noisy measurements are a cause for poorer results, for example in the correlations. You have also used the different impact of clouds as an explanation of some differences. I am not convinced that these explanations are well enough proven, at least by what I see in this manuscript. When you have monthly means, most of the results will have very small standard errors in the means, and noise itself becomes much less of an issue. This can be demonstrated for each instrument, and some will be more "noisy" than others, this is still true, depending on the number of data points (and the actual single-profile noise). I would urge you to more carefully consider those comments and convince yourselves of certain cases where these statements are made. If you are not convinced, or convincing, maybe you should invoke unknown systematic effects as well, as another option that could also*

*play a role (when one wants to try to provide a range of options for some differences without a more complete investigation, which could take a while for the multitude of data sets you are considering here, I agree). Alternatively, if you do have a few good examples, you could add supplementary material to show differences of time series with the noise values (for example), or something else relating to cloud studies (or another reference, possibly). I also tend to think that sampling will play a bigger role than one might think in some of the differences (for example in the issues mentioned in the paragraph before Section 4.4), even if you already do mention some examples of this issue. I just want to avoid statements that sound like careful work has gone into the conclusion, with little proof given; while there can be (is) some level of trust regarding work not shown in the paper, I am not entirely convinced that all the possible explanations have been vetted enough. One can try to investigate and comment about a few of the more obvious cases with poorest results, for example, at certain pressures or latitudes. It is also somewhat surprising to see some of the differences between the various MIPAS retrievals, in terms of how some of the larger differences can come about. Again, the main flavor of the results will most likely not change, and it would be a large undertaking to try to resolve "everything", so I (and others) will need to read this as the way things are, for now, not that one shouldn't expect better agreement in the end, given enough time, etc... but there are still a number of unresolved issues. Some of the writing gives a lot of description of differences, with maybe not enough "potential explanations", and we can infer that some things are not understood yet, which is not completely unexpected either. This does not make this work unworthy of publication, in my view, although one would prefer to see a lot of these uncertainties reduced, or somewhat better explained.*

**It is correct that due to the fact that we use monthly means we cannot talk about "noise" in the classical sense. In fact, monthly means are only considered if they are larger than the corresponding standard error, so that this component is more or less irrelevant. At three occasions in the manuscript we mention "noise". In these three occasions we really refer to "noise". The large variability**

we see from month to month in some data sets was denoted as "scatter" in our manuscript to differentiate between the classical "noise" and the "noisy" behaviour we see in the monthly mean data. A thorough discussion on the reasons for differences between the data sets is given in Lossow et al. (2017). From this study we know that differences in sampling, cloud influence or NLTE have an influence. Additionally, there are also systematic differences for example from calibration problems. However, for the time series comparison we would rank sampling biases as well as systematic errors as the most important reason for differences as was discussed by Toohey et al. (2013) based on trace gas climatologies. We added the following text to the "Summary and Conclusion" section: *There are multiple reasons that give rise to the observed differences between the individual data sets. A thorough discussion on this is given in Lossow et al. (2017). From this study we know that the most important contributions arise from differences in temporal and spatial sampling, the influence of clouds or NLTE effects. Other reasons include systematic differences, for example calibration problems. However, for the time series comparison we would rank sampling biases as well as systemtatic errors as the most important reason for the differences as was discussed by Toohey et al. (2013) based on trace gas climatologies.*

*Minor Comments*
*- pg. 2, L3, change ratio to ratios.*
**Done.**

*- pg. 2, L8, change :when data" to "if data", although my specific comments are not that positive regarding this sort of statement.*
**Done. See also our answer to the specific comments.**

*- pg. 3, L18; I would suggest something shorter/better like "with water vapour*

*abundances recovering after 2004-2005".*
**We changed the sentence as follows: "with water vapour abundances starting
to recover from 2004–2005 onwards."**

*- pg. 4, Eq. (1), interesting use of "z" for pressure, rather than "p", but this is
just a personal preference, nothing to really change.*
**It is correct, that in our analyses the altitude coordinate is pressure, therefore
it should correctly read "p(z)". However, with simply using "z" in the equation
we indicate the dependence on altitude which can either be altitude or pressure
altitude.**

*- pg. 5, lines 8 and 11, there is a change in tense (from "was calculated" to
"are discarded"); I would recommend using the same tense in general, inasmuch as
possible (e.g., "were discarded").*
**This has been corrected.**

*- pg. 5, L15, maybe change "refined" to "relaxed", or "less stringent criterion".*
**We changed "refined" to "relaxed".**

*- pg. 6, L11, typo in "alltogether" [altogether].*
*- pg. 6, L12, change "ratio" to "ratios".*
*- pg. 6, L22, delete "the" before "the".*
**These typos have been corrected.**

*- pg. 6, L26, change "was 3.6" to "removed 3.6".*
**We changed the sentence as follows: *For the tropical and the mid-latitude bands
3.6% and 3.7%, respectively, of the data were removed.****

*- pg. 6, L29, add a comma after "measurements".*

*- pg. 7, L1, change "result" to "results".*
*- pg. 9, L18, are compared qualitatively.*
*- pg. 10, L26, delete "and" before "2011".*
*- pg. 11, L14, available for these altitude and latitude regions.*
*- pg. 11, L18, change "as those" to "than those".*
*- pg. 11, L33, change "in order" to "on order".*
*- pg. 12, L14-15, in the other data sets, anomalies up to only 0.4-0.8...*
*- pg. 12, L17, anti-correlated with the time series*
*- pg. 12, L23, add a comma after V5H.*
*- pg. 13, L1, are found during 2004-2008, when SAGE II...*
*- pg. 13, L9, change "regions" to "region".*
*- pg. 13, L17, change slightly decreasing to decreasing slightly.*
**These issues have been corrected.**

*- pg. 13, L21, 22, there is a lot of repetition of the same references to the drops in $H_2O$.*
**The references on P13, L21.22 have been removed.**

*- pg. 14, L17, change "coefficient" to "coefficients", and one line lower, change "is" to 'are".*
*- pg. 14, L20, change "in form" to "in the form" or to "as".*
*- pg. 14, L30, change "NOM than" to "NOM as".*
*- pg. 14, L31, most data sets between 1 and 30 hPa.*
*- pg. 15, L9, change "varies" to "vary".*
*- pg. 15, L19, "the number of months of overlap between time series is given..."*
*- pg. 15, L25, the number of overlap months is not that high...*
*- pg. 16, L2, the number of overlap months is rather low (same for L5/6, and L10, and L20 on pg. 17).*
*- pg. 16, L7, overview of the temporal...*

*- pg. 16, L11, An example of a negative correlation, despite a high number of overlapping months, is for the correlation ...*
**All suggested corrections/changes have been considered.**

*- Also on this page, it is not very surprising when ACE-FTS data versions or MIPAS versions correlate well... it is the opposite that is more surprising.*
**In case of ACE-FTS a high correlation may not be surprising, but for the different MIPAS data sets this is not necessarily given since there are so many differences in these data sets. What we simply do here is to describe the correlations which are in the correlation matrices most pronounced visible, irrespective if this is an expected or not expected result.**

*- pg. 16, L23, which "both quantities" are you talking about here, please clarify.*
**With both quantities we mean the number of overlap months and the correlation coefficient. We changed the sentence as follows:** *Therefore, for assessing the agreement between two data sets both quantities, the number of overlap months and the correlation coefficient, should be taken into account.*

*- pg. 17, L14, change "were both data sets" to "for which both data sets".*
*- pg. 17, L19, change "at the" to "on the" (x-axis and y-axis).*
*- pg. 18, L17, drifts are significant in most cases.*
*- pg. 18, L26, change "pattern" to "patterns" [are...].*
*- pg. 18, L34, Exceptions are HIRDLS... and MAESTRO...*
*- pg. 19, L25, change "because" to "that".*
*- pg. 19, L27, influenced differently by clouds.*
**These issues have been corrected.**

*- pg. 19, L30/31, Larger deviations in the lower mesosphere occur in the case*

*of the MIPAS NOM data sets, which are close to their upper retrieval limit there, and thus more uncertain.*
**We changed the sentence as suggested.**

*- pg. 20, Since the dehydration is only partly a seasonal oscillation,...*
**We changed this text part as follows to be more precise what the point is:**
***Since the dehydration is more a seasonal phenomenon, and accordingly is less characterised by a sinusoidal behaviour, the usage of sinusoidal functions for the de-seasonalisation is not the optimal choice. Instead, the average approach (see Sect. 3.1) would be the more adequate choice for the de-seasonalisation in this region.***

*- pg. 20, L8, were also assessed; this indicates if the longer-term variations (trends) ...*
**"drifts" is correct here, since our study is about "drifts" and not "trends". Further, we think it is not necessary to split the sentence.**

*- pg. 20, L11, change "level" to "levels".*
**Done.**

*- pg. 20, L11, The most significant drifts were found in the tropics, where there is low..., which...*
**We changed the sentence as follows:** ***The majority of significant drifts were found in the tropics (the latitude region with the lowest spread/variability), which makes drift detection considerably easier.***

*- pg. 20, L13, Drifts were also calculated...*
**We changed the sentence as follows:** ***The same drift approach as used here has been used by Lossow et al. (2018b) to calculate drifts from profile-to profile***

*comparisons (using coincident data).*

*- pg. 20, L22, delete "amongst others".*
**We deleted "among others" as requested.**

*- pg. 20, L32, that in addition to correcting...*
*- pg. 20, L33, changes in the calibration were made within the HIRDLS mission...*
**These changes have been implemented.**

*- pg. 20, L34, data sets encounter large uncertainty...*
**The sentence has been changed as follows:** *The MAESTRO data set encounters large uncertainty (noise) at 80 hPa (in the correlations and drifts) which is related to the vicinity to the uppermost limit of these retrievals.*

*- References, there are a few refs. with no doi numbers (Randel, Remsberg, von Clarmann).*
**Missing dois have been added.**

*- pg. 28, bottom line, and the spread are more easily compared.*
**Done.**

*- Figure 5, one could also find an rms diagnostic versus pressure and contract the time axis this way, as an additional Figure that could overlay the three latitude regions, so as to get a maybe better overview of this quantity over pressure and latitude (and some of the colors are not so easy to differentiate).*
**If we do understand this comment correctly, the referee means that we should choose a representation here that is no longer dependent on time. We do not agree with this suggestion because the variability of the spread over time is an important information.**

*Figure 10, last sentence, The second number indicates the number of months for which both data sets ...*

*- Figure 11, line 2, change "at" to "on" (x-axis, y-axis). Line 4, upper left the overall overlap period between data sets is given. The second number indicates for how many months ...*

**These sentences have been corrected.**

**References**

**Lossow et al., The SPARC water vapour assessment II: comparison of annual, semi-annual and quasi-biennial variations in stratospheric and lower mesospheric water vapour observed from satellites, Atmos. Meas. Tech., 10, 1111 – 1137, https://doi.org/10.5194/amt-10-1111-2017, 2017.**

**Lossow et al., Trend differences in lower stratospheric water vapour between Boulder and the zonal mean and their role in understanding fundamental observational discrepancies, accepted for publication in ACP, 2018a.**

**Lossow et al., The SPARC water vapour assessment II: Profile-to-profile comparisons of stratospheric and lower mesospheric water vapour data sets obtained from satellite, in preparation, 2018b.**

**Toohey et al., Characterizing sampling biases in the trace gas climatologies of the SPARC Data Initiative, J. Geophys. Res., 118, 11847 – 11862, https://doi.org/10.1002/jgrd.50874, 2013.**

**Walker et al., The SPARC water vapour assessment II: Data set overview, in preparation.**

[Figure]

[Figure]

**Fig. 1.** The drifts (left) and corresponding signifcance level (right) between the de-seasonalised time series of the SMR 489 GHz data set and the other data sets for the three latitude bands considered.

---

## Author Comment (AC2) · 25 Jun 2018

We thank Hugh Pumphrey for the constructive, helpful criticism and the suggestion for revision. We have revised the manuscript accordingly.

*General comments*
*This is a useful summary paper, which, for the most part, presents a large and complex body of information in a digestible manner. Most of the items that make it difficult for the reader are related to the large number of datasets from a single instrument*

[Figure]

*(MIPAS). If it were possible to do anything to separate the inter-MIPAS comparisons from the (more important) comparisons between different instruments, then I would like to see this done. But I recognise that this might be too large a change to be made. Like referee 1, I am conscious that this paper does not provide any kind of guidance to the reader as to which data sets are the most useful. I imagine that this is deliberate and is done to avoid annoying any of the data providers. I nevertheless feel that some sort of opinion as to which datasets are the most useful for which purposes would not be out of place.*

**We can understand that the huge number of MIPAS data sets is somewhat overwhelming. However, since these data sets exists, they also have a right to be assessed. These data stem from 4 different processors and there are a lot of differences between the data sets as shown in our paper as well as in the other WAVAS papers. The intention of WAVAS is to provide a full assessment of "all" available stratospheric data sets. Plenty of time series analyses and assessments using less data sets can be found elsewhere (e.g. Hegglin et al, 2013; Hegglin et al., 2014; Khosrawi et al., 2016; Weigel et al., 2016; Noel et al., 2018; Lossow et al., 2018). We included now an paragraph in the manuscript on the differences between the MIPAS data sets (see our answer below to the specific comment on Figures 2-4). We agree that it would be good to give some guidance on which data set to use for further studies. However, this is quite difficult to judge since this decision depends on the scientific application and on which altitude region or latitude region the study is focused on. For trend analyses the longest data sets with the highest spatial and temporal coverage have of course for such studies a clear advantage. We changed the last paragraph of the conclusion as follows and hope that this will give at least some guidance:** *Nevertheless, although the water vapour data sets have been thoroughly assessed in this study it is difficult or rather impossible to judge on which data set is the best one to use for future modelling and observational studies. This simply can only be answered with respect to the*

*specific science application the data set should be used for. For future studies on e.g. water vapour trends we can state that the data sets that provide the longest measurement record with a high spatial and temporal coverage have an advantage over the ones which provide only observations in specific latitude bands and/or altitude regions. For data sets that have a drift relative to other data sets as e.g. SMR 489 GHz, the drift has to be taken into account and data sets that are simply too short (less than one year) as e.g. ILAS-II and SMILES cannot be used for trend studies at all.*

*Specific comments*

- *Page 4 line 10: The authors note that the data from UARS MLS are not considered. I do not think these data would add much as there are less than 18 months worth. But the authors have included the ILAS-II and SMILES data, which cover even shorter time periods, so I think they should explain why they are including ILAS-II and SMILES, but not including UARS MLS. (Disclaimer: I am responsible for the UARS MLS water vapour data.)*
  **The aim of WAVAS II was to include all data sets that performed observations in the period from 2000 to 2014 (or extended to 2016 as it is done in some other WAVAS II papers). To our knowledge UARS/MLS $H_2O$ measurements ceased in 1993 and that only measurements from the other trace gases are available until 2001. An assessment of the pre-2000 data sets was done within the first WAVAS project and can be found in the SPARC WAVAS Report published in 2000.**

- *Figure 1: The labelling of the colour bar is rather cluttered; it might be preferable to label only 2,3,4,5,6,7, and 8 ppmv.*
  **We agree and changed the labeling of Figure 1 as suggested.**

- *Page 8 line 14: I would remove the words "(contour time series)" as the data are*

*presented as an image — no contours have been drawn.*

**It is correct that we have not drawn any contour lines. However, the definition of a contour plot is as follows: "A contour plot is a graphical technique for representing a 3-dimensional surface by plotting constant z slices, called contours, on a 2-dimensional format. That is, given a value for z, lines are drawn for connecting the (x,y) coordinates where that z value occurs." This exactly what we are doing, but instead of using lines we use filling of the contours. Thus, it correctly should read "filled contour". However, this is detail is not very useful and somehow we need to distinguish our time series plots from each other and thus we would prefer to keep the header using the term "contour".**

- *Page 8 line 27: Again (and in several subsequent places, including in the supplement), remove the word "contour" as figures 1, 5, and S1-S3 contain no contours.* **As stated in our answer above, these are nevertheless contour plots, but without explicitly plotting contour lines. Thus, we would rather keep the word "contour" in the text and figures to differentiate these figures from the other time series plot where we consider the time series on specific pressure levels.**

- *Figure 2-4: I do rather wish that the various teams involved with MIPAS would agree on one best product. Half of the products shown in these figures are from this one instrument. I understand that the instrument has various operating modes which are not directly comparable, so a single product may not be practical. But 13 different products are very confusing for the reader and the data user. It might have been preferable to first form some sort of combined or approved MIPAS dataset (or, at most, one for each operating mode) to be compared to other instruments. I do not imagine that the authors will want to re-design the entire paper along these lines. But for the purposes of these figures it might be better to show only one MIPAS dataset (and possibly, only one ACE-FTS data set —*

*why do we need V2.2 if V3.5 is supposed to be an improvement?).*

**We can really understand this point of criticism since at a first glance including all 13 MIPAS data sets looks a bit like an overkill. To simply pick one data set (or a selection of data sets) from MIPAS is not possible due to the differences between these data sets (due to usage of different micro-windows, different retrieval choices etc). We have to apologize here that we completely missed out to motivate in the manuscript why we want/need to include all MIPAS data sets in this comparison. Therefore, we included in Section 2 a similar paragraph as the one given in Nedoluha et al. (2017) on the differences of the MIPAS data sets:** *This especially holds for MIPAS where 13 data sets have been included in this comparison. The MIPAS measurements are processed by four different processing centers: (1) the University of Bologna (Dinelli et al., 2010), (2) the European Space Agency (ESA; Raspollini et al., 2013), (3) IMK/IAA (von Clarmann et al., 2009; Stiller et al. 2012), and (4) Oxford (Payne et al., 2007). The four processors differ in several respects, such as their choices of spectral ranges (so called micro-windows), the vertical grid on which the retrievals are performed (pressure or geometric altitude), the choice of regularization (and related to this, the vertical resolution), the choice of spectroscopic database, the sophistication of the radiative transfer (in particular, whether or not non-LTE emissions are considered), and whether or not any attempt is made to account for horizontal inhomogeneities, and the a priori and the assumed p-T profile. Indeed, the temperature used might be a large source of error for species retieved in LTE regions. Some of the different processing schemes also make use of different level-1b data versions (here V5 and V7) based on different ESA calibrations. The spread of results seen for MIPAS indicates how specific choices within a retrieval approach may influence the retrieval results.* **Selecting one specific MIPAS data set (the best one, obviously) might rather be an outcome of this study but not an input. Re-**

**garding the two ACE-FTS versions that are included in this assessment: We had an open data set policy to represent a database as complete as possible. All data sets were allowed to participate. The ACE-FTS team wished to include both data sets, v2.2 (well validated) and v3.5 (not really validated, covering a longer time period).**

- *Page 10-12: Many of the features of the data described here are rather hard to see in figures 2-4, on account of the large number of lines. I am not sure what to suggest (other than not showing all the MIPAS data!).*
  **We agree that with such a high number of data sets the features we described here becomes hard to see. However, this is the drawback of performing a multi-dataset assessment. For the sake of completeness it is important to have all data sets included. Nevertheless, by just zooming into the figures we managed to see these features despite the high number of instruments. Nowadays, many scientist anyway read papers rather on the computer screen than printing them out. Separating the MIPAS data sets from the other data sets is no solution since then we would not be able to include MIPAS into the comparison (since picking one data set is also no option as discussed above). Comparisons of water vapour time series using less data sets are published elsewhere (e.g. Hegglin et al., 2013; Hegglin et al., 2014; Weigel et al., 2016; Khosrawi et al., 2016; Noel et al. (2018); Lossow et al., 2018). Further, there actually has been some optimisation in the plotting sequence of the time series with the aim to benefit the sparse data sets. Furthermore, the time series analyses shown Figs. 2-4 provides only a qualitative assessment. From these figures we learn more on the characteristics of the data sets when we look at the outliers instead of on the data sets that agree well with each other. Further, the more important results in this study are the assessment of the correlation and drifts where we overcome the problem of the huge amount**

**of data sets by using the matrix plots and giving quantitative estimates of the differences.**

- *Figure 5: The black dots are very difficult to see, especially against the darker end of the colour scale. Potential solutions include joining the dots with a line and/or using a colour (red?) which does not form part of the colour scale.*
  **Thanks a lot for the suggestion. We increased the size of the dots and changed the colour from black to red.**

- *Page 14 lines 1-15: One of the most striking features of the figure is the change in 2012 caused by the end of the Envisat mission and hence of the myriad MI-PAS datasets. It strikes me that the use of the max-min difference to quantify spread means that this plot mostly tells you about where the noisiest dataset is at its noisiest. I have to question whether this is the most useful measure of either atmospheric variability or overall data quality.*
  **We have tested several methods to calculate the spread and derived qualitatively the same results. We prefer the spread calculation using the max-min differences since it makes the spread calculation most comprehensible and shows most clearly that the largest spread between the data sets is found where the largest variability in H$_2$O is found, in agreement with what was found in Lossow et al. (2017). Further, it should be noted that a pre-screening has been performed to remove outliers and to get reasonable estimates. It is correct that a striking feature is the change in 2012 due to the end of the Envisat mission. However, keeping these two years in the figure is worth since it quite clearly shows that with a few data sets the spread is decreasing, but the characteristic features (largest spread found in the areas of largest variability) are not that pronounced any longer. Thus, showing that a few data sets are not sufficient to get a good statistic.**

- *Figures 7-9: These figures are an interesting way of showing a large amount of summary information in a clear way, and in a small space. Something that caused me a bit of confusion was the way that the numbers in the upper triangle do not always align with those in the lower triangle. This is because different levels have different datasets available. It might be worth inserting blank rows into one or other triangle in each pane so that the two triangles have the same numbering scheme.*
**We agree, but we have to keep these gaps to save space, because otherwise the boxes get even smaller as they already are.**

- *Figures 11-13: In addition to the suggestion I make regarding figures 7-9, figures 11-13 have text on them which is VERY small. It is commonly recommended that text on a figure should be no smaller than the figure caption text in the final typeset version of the article. There is clearly a bit of leeway on this recommendation, but the text on this figure is so tiny that it is very annoying for the reader, especially for middle-aged readers who are still cross that they need reading glasses. I am not sure what to suggest here, because simply making the text bigger will not work: in some cases it is already impinging on the diagonal lines.*
**We agree that the numbers in the boxes are really hard to read. However, we really tried to find a solution for this problem, but could not come up with a better idea. Nevertheless, the numbers are additional information and the most important information in this figure is the drift given by the colours and the colour bar as well as if the drift is significant or not by the green boxes and a slash. For reading the numbers one in fact has to use the pdf and zoom in. However, to make it easier for the readers who prefer a paper version, we shortened the caption of Fig. 11 so that the size of these figures is now a bit increased and added these three figures to the supplement where we can provide them in a even larger size than in the manuscript.**

[Figure]

- *Page 21: dedication. I too have good memories of working briefly with Jo Urban, and was saddened to hear of his passing at such a young age.*

  *Technical corrections*

- *Page 2 line 1: "allowed considering the time period" reads rather oddly. Maybe write "allowed us to consider the time period" or "allowed the consideration of the time period".*
  **We have changed this phrase as suggested.**

- *Page 3 line 17: "One drop (also known as the millennium drop) . . . " The "also" does not read right as you have not first given another name by which the drop is known. Maybe write "One drop (sometimes known as the millennium drop) . . . ".*
  **We changed the sentence as follows: *One drop (sometimes denoted as the millennium drop) occurred in 2000..........***

- *Page 11 Line 25: remove comma after "Both"*
  **Done.**

- *Page 14 line 30: replace "than" with "as"*
  **Done.**